# Structural basis for the activation of PLC-γ isozymes by phosphorylation and cancer-associated mutations

Nicole Hajicek[1], Nicholas C Keith[1], Edhriz Siraliev-Perez[2], Brenda RS Temple[2,3], Weigang Huang[4], Qisheng Zhang[1,4,5], T Kendall Harden[1], John Sondek[1,2,5]*

[1]Department of Pharmacology, The University of North Carolina at Chapel Hill, Chapel Hill, United States; [2]Department of Biochemistry and Biophysics, The University of North Carolina at Chapel Hill, Chapel Hill, United States; [3]R L Juliano Structural Bioinformatics Core Facility, The University of North Carolina at Chapel Hill, Chapel Hill, United States; [4]Division of Chemical Biology and Medicinal Chemistry, The University of North Carolina at Chapel Hill, Chapel Hill, United States; [5]Lineberger Comprehensive Cancer Center, The University of North Carolina at Chapel Hill, Chapel Hill, United States

**Abstract** Direct activation of the human phospholipase C-γ isozymes (PLC-γ1, -γ2) by tyrosine phosphorylation is fundamental to the control of diverse biological processes, including chemotaxis, platelet aggregation, and adaptive immunity. In turn, aberrant activation of PLC-γ1 and PLC-γ2 is implicated in inflammation, autoimmunity, and cancer. Although structures of isolated domains from PLC-γ isozymes are available, these structures are insufficient to define how release of basal autoinhibition is coupled to phosphorylation-dependent enzyme activation. Here, we describe the first high-resolution structure of a full-length PLC-γ isozyme and use it to underpin a detailed model of their membrane-dependent regulation. Notably, an interlinked set of regulatory domains integrates basal autoinhibition, tyrosine kinase engagement, and additional scaffolding functions with the phosphorylation-dependent, allosteric control of phospholipase activation. The model also explains why mutant forms of the PLC-γ isozymes found in several cancers have a wide spectrum of activities, and highlights how these activities are tuned during disease.

*For correspondence:
sondek@med.unc.edu

Reviewing editor: Neel Shah,

## Introduction

The 13 phospholipase C (PLC) isozymes expressed in humans preferentially hydrolyze the membrane phospholipid phosphatidylinositol 4,5-bisphosphate ($PIP_2$) to generate the second messengers diacylglycerol and inositol 1,4,5-trisphosphate ($IP_3$) (*Harden and Sondek, 2006*; *Kadamur and Ross, 2013*). Diacylglycerol is retained within membranes where it recruits and activates numerous proteins including conventional isoforms of protein kinase C. In contrast, $IP_3$ diffuses throughout the cytosol where it binds to $IP_3$ receptors embedded in endoplasmic reticulum leading to mobilization of sequestered calcium. PLC-mediated depletion of $PIP_2$ also modulates the activities of several ion channels and proteins with phosphoinositide-binding domains. Thus, the PLCs coordinate fluctuations in $PIP_2$ levels and the bifurcating signaling pathways emanating from $PIP_2$ hydrolysis to regulate numerous cellular processes, including fertilization and embryogenesis, cell proliferation and differentiation, as well as various types of cell migration (*Asokan et al., 2014*; *de Gorter et al., 2007*; *Jones et al., 2005*; *Mouneimne et al., 2004*).

The two PLC-γ isozymes, PLC-γ1 and PLC-γ2, are unique among the PLCs in that they are directly activated by tyrosine phosphorylation. PLC-γ1 and PLC-γ2 are phosphorylated on equivalent sites, Tyr783 and Tyr759, respectively, and this phosphorylation is typically required to stimulate

**eLife digest** Many enzymes are poised to receive signals from the surrounding environment and translate them into responses inside the cell. One such enzyme is phospholipase C-γ1 (PLC-γ1), which controls how cells grow, divide and migrate.

When activating signals are absent, PLC-γ1 usually inhibits its own activity, a mechanism called autoinhibition. This prevents the enzyme from binding to its targets, which are fat molecules known as lipids. When activating signals are present, a phosphate group serves as a 'chemical tag' and is added onto PLC-γ1, allowing the enzyme to bind to lipids.

Failure in the regulation of PLC-γ1 or other closely related enzymes may lead to conditions such as cancer, arthritis and Alzheimer's disease. However, it remains unclear how autoinhibition suppresses the activity of the enzyme, and how it is stopped by the addition of the phosphate group.

Here, Hajicek et al. determine in great detail the three-dimensional structure of the autoinhibited form of the enzyme using a method known as X-ray crystallography. This reveals that PLC-γ1 has two major lobes: one contains the active site that modifies lipids, and the other sits on top of the active site to prevent lipids from reaching it. The findings suggest that when the phosphate group attaches to PLC-γ1, it triggers a large shape change that shifts the second lobe away from the active site to allow lipids to bind.

The three-dimensional structure also helps to understand how mutations identified in certain cancers may activate PLC-γ1. In particular, these mutations disrupt the interactions between elements that usually hold the two lobes together, causing the enzyme to activate more easily.

The work by Hajicek et al. provides a framework to understand how cells control PLC-γ1. It is a first step toward designing new drugs that alter the activity of this enzyme, which may ultimately be useful to treat cancer and other diseases.

phospholipase activity. Several classes of tyrosine kinases phosphorylate and activate the PLC-γ isozymes. These include a large number of receptor tyrosine kinases (RTKs) including Trk receptors (*Minichiello et al., 2002*; *Vetter et al., 1991*) and many growth factor receptors such as epidermal growth factor receptor (EGFR) (*Kim et al., 1991*; *Nishibe et al., 1990*; *Peters et al., 1992*; *Takahashi et al., 2001*; *Wahl et al., 1989*). A second large group is soluble tyrosine kinases coupled to immune receptors and includes members of the Src, Syk, and Tec families (*Law et al., 1996*; *Nakanishi et al., 1993*; *Schaeffer et al., 1999*). In this way, the PLC-γ isozymes are poised to transduce signals initiated by a wide variety of extracellular stimuli.

The regulated, phosphorylation-dependent activation of PLC-γ1 and -γ2 controls numerous aspects of biology including proper development of the vascular, neuronal, and immune systems during embryonic development, adaptive immune responses, neuronal transmission, bone homeostasis, chemotaxis, and platelet aggregation (*Yang et al., 2012*). The PLC-γ isozymes have also recently emerged as drivers of several human diseases (*Koss et al., 2014*). Notably, genome-wide sequencing studies have demonstrated that PLC-γ1 and PLC-γ2 are frequently and recurrently mutated in several leukemias and lymphomas. In fact, PLC-γ1 is the most frequently (~40%) mutated gene in adult T cell leukemia/lymphoma, where mutant forms of the isozyme are presumed to drive oncogenesis through enhanced phospholipase activity coupled to elevated NFAT- and NF-κB-dependent transcription (*Kataoka et al., 2015*; *Vaqué et al., 2014*). Moreover, activating mutations in PLC-γ2 arise with high frequency (~30%) in patients with B cell leukemias treated with ibrutinib (*Woyach et al., 2014*), a covalent inhibitor of Bruton's tyrosine kinase (BTK). PLC-γ2 is a major BTK substrate and mutations in PLC-γ2 likely function as escape mutations, reactivating pathways controlling cell survival and proliferation that are usually rendered quiescent by ibrutinib treatment. Mutated, and presumably active, forms of the PLC-γ isozymes are also found in patients with angiosarcomas (*Behjati et al., 2014*), and several disorders associated with dysregulated immune responses (*Ombrello et al., 2012*; *Zhou et al., 2012*), inflammatory bowel disease (*de Lange et al., 2017*), and familial steroid-sensitive nephrotic syndrome (*Gbadegesin et al., 2015*; *Parker et al., 2019*). A naturally occurring variant of PLC-γ2 that is moderately active is strongly associated with protection from late-onset Alzheimer's disease (*Magno et al., 2019*; *Sims et al., 2017*) and

highlights the notion that aberrant PLC activity can be either deleterious or beneficial depending on the context. However, for the vast majority of mutant forms of PLC-γ1 and PLC-γ2, an increase in lipase activity has not been demonstrated directly.

While the PLC-γ isozymes control essential aspects of both normal and disease-associated cellular processes, the molecular mechanisms controlling these enzymes remain elusive. Broadly, phosphory-lation-dependent activation of PLC-γ1 and PLC-γ2 is controlled by an array of regulatory domains unique to these isozymes. In particular, the regulatory array harbors the obligatory sites of tyrosine phosphorylation (*Gresset et al., 2010*; *Kim et al., 1991*; *Nishibe et al., 1990*), and also includes an SH2 domain (nSH2) required for tyrosine kinase binding (*Bae et al., 2009*). The array also mediates basal autoinhibition of phospholipase activity, since removal or mutation of a second SH2 domain (cSH2) within the array results in robust and constitutive activation of PLC-γ isozymes in vitro and in cells (*Gresset et al., 2010*; *Hajicek et al., 2013*). The regulatory array is completed by a split PH (sPH) domain and an SH3 domain, which modulate PLC activity by scaffolding numerous signaling and adaptor proteins (*Sherman et al., 2016*; *Walliser et al., 2008*).

Although these general aspects of the regulation of PLC-γ1 and PLC-γ2 are well-documented, several fundamental questions remain unresolved. For example, the other half of the autoinhibitory interface formed by the cSH2 domain has not been identified, and how this domain enforces autoin-hibition of phospholipase activity is unknown. Possibilities include physical occlusion of the lipase active site, steric hindrance that prevents the enzyme from engaging membranes, or a combination of both as is observed for the PLC-β isozymes (*Hicks et al., 2008*). Although the cSH2 domain is pre-sumed to be the primary arbiter of autoinhibition, several reports have also implicated the sPH domain in maintaining an autoinhibited state (*Everett et al., 2011*; *Gresset et al., 2010*). How this domain might contribute to autoinhibition is unknown.

Similarly unclear is how autoinhibition is relieved by tyrosine phosphorylation. Phosphorylated Tyr783 in PLC-γ1 binds with nanomolar affinity to the cSH2 domain, and this engagement presum-ably couples a large, but ill-defined conformational rearrangement within the array to activation (*Gresset et al., 2010*; *Hajicek et al., 2013*; *Poulin et al., 2005*). However, even these few mechanis-tic details are under debate since an alternative model posits engagement of RTKs by the cSH2 domain as an initial step required for activation (*Huang et al., 2016*).

The paucity of mechanistic understanding of the activation of the PLC-γ isozymes is largely attrib-utable to a lack of structural information. While structures of isolated portions of the regulatory array of PLC-γ1 are available, these structures provide an incomplete and sometimes erroneous context for defining how the array integrates the functions of basal autoinhibition and phosphorylation-dependent activation. There are no structures of full-length PLC-γ isozymes.

Here, we describe the 2.5 Å-resolution crystal structure of essentially full-length PLC-γ1. The structure highlights a regulatory array exquisitely positioned to prevent membrane engagement of the catalytic core while simultaneously arranged to scaffold tyrosine kinases and additional regula-tory proteins into a signaling nexus. Kinases have unfettered access to the nSH2 domain of the regu-latory array and will dock here initially. In contrast, the phosphotyrosine binding pocket within the cSH2 domain is buried through interactions with the catalytic core. Docked kinases are well posi-tioned to phosphorylate Tyr783 located within a nearby solvent-exposed loop and phosphorylated Tyr783 (pTyr783) is expected to bind the cSH2 domain to disrupt its interaction with the catalytic core. This disruption is predicted to favor a substantial rearrangement of PLC-γ1 required for pro-ductive membrane engagement and hydrolysis of PIP$_2$. The structure also explains how cancer-asso-ciated substitutions disrupt autoinhibition to elevate basal PLC-γ1 activity and contribute to supra-activation of the isozyme in the context of receptor overexpression. The combined effects of substi-tution and tyrosine phosphorylation suggest that cellular context, for example overexpression of EGFR found in many cancers, will be especially important for understanding diseases modulated by the PLC-γ isozymes.

## Results

### Structure of full-length PLC-γ1

In addition to the aforementioned array of regulatory domains, the PLC-γ isozymes also possess a set of core domains common to most other isoforms of PLC: an N-terminal PH domain, two pairs of

EF hands, a catalytic TIM barrel, and a C2 domain. The regulatory array bisects the TIM barrel,

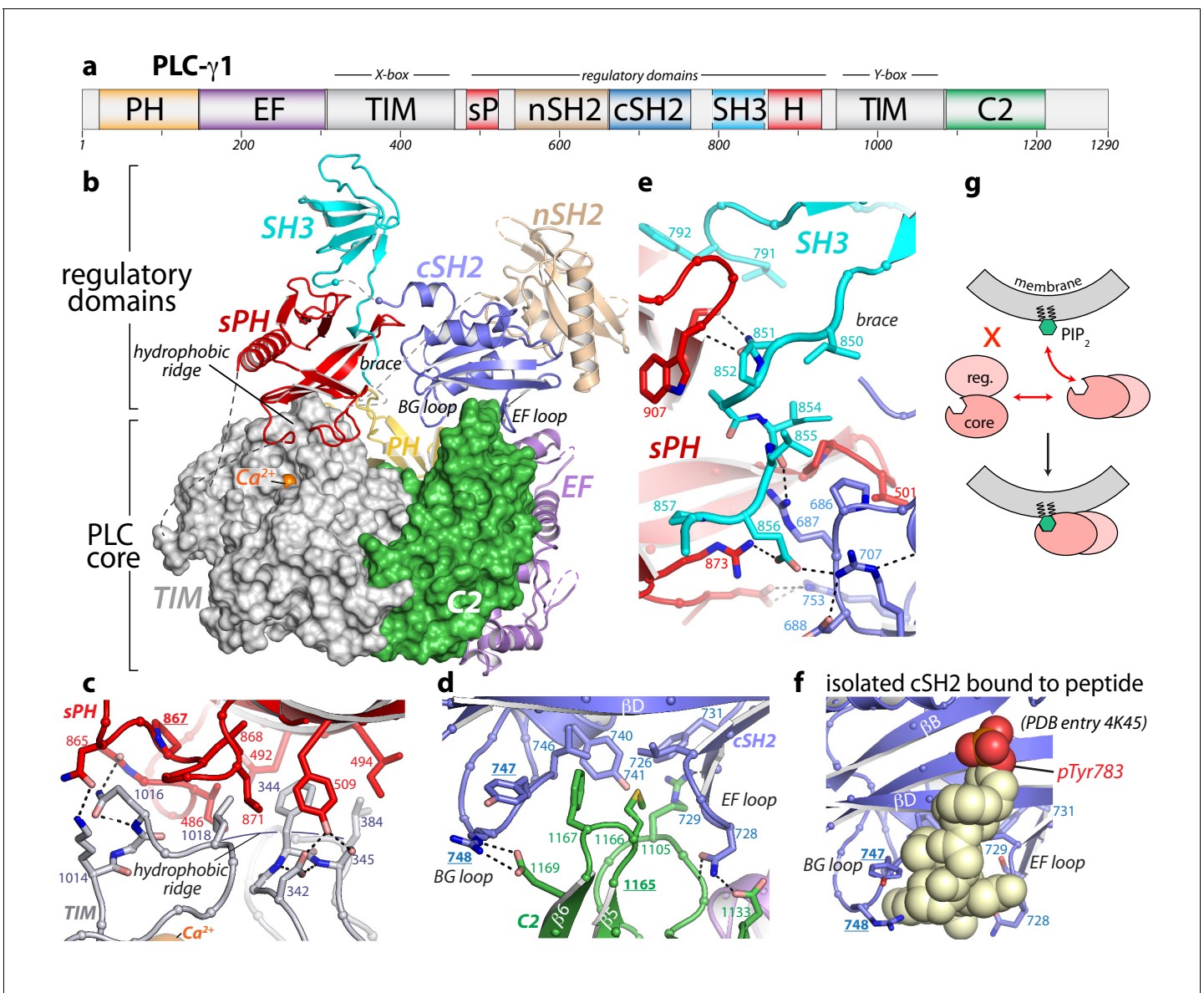

**Figure 1.** Crystal structure of autoinhibited PLC-γ1. (a) Domain architecture of PLC-γ1 drawn to scale. (b) 2.5 Å resolution structure of PLC-γ1, domains are colored as in (a); TIM barrel and C2 domain are depicted as surfaces to highlight interactions with regulatory domains. The calcium cofactor (orange sphere) marks the active site and dashed lines indicate regions not built due to the absence of observable electron density. Borders of the Δ25 deletion (residues 766–790) used to facilitate crystallization are indicated with spheres. The hydrophobic ridge of the TIM barrel, which interacts with lipid membranes to facilitate catalysis, is occluded by the sPH domain. This arrangement of the sPH domain is supported by contacts with the cSH2 domain and further reinforced by a 'brace' formed by the C-terminal extension of the SH3 domain. (c–d) Structural details between the regulatory and core domains of PLC-γ1. (e) Expanded view of the SH3 domain. (f) Structure of the isolated cSH2 domain of PLC-γ1 bound to a peptide (spheres) encompassing phosphorylated Tyr783 (red) of PLC-γ1. Orientation is approximately the same as in panels (b) and (d). For panels (c–f) interfacial residues are numbered, dashed lines are hydrogen bonds, and residues mutated in *Figure 3* are underlined. (g) Schematic emphasizing large conformational change that is proposed to occur before PLC-γ1 can access membrane-resident PIP₂.

The online version of this article includes the following figure supplement(s) for figure 1:

**Figure supplement 1.** Primary sequence alignment of PLC-γ isozymes.

**Figure supplement 2.** Crystallographic-grade PLC-γ1(21–1215) Δ25 is fully autoinhibited in cells and in vitro.

**Figure supplement 3.** The surface of autoinhibited PLC-γ1 is electronegative and unlikely to bind biological membranes.

**Figure supplement 4.** Structural comparisons between autoinhibited PLC-γ1 and fragments of the PLC-γ isozymes.

subdividing this domain into the X- and Y-boxes (*Figure 1a*, *Figure 1—figure supplement 1*). To facilitate crystallization, several regions predicted to be disordered were removed from the construct used for structure determination. In particular, 20 and 75 residues were deleted from the N- and C-terminus, respectively. In addition, an internal loop of 25 residues connecting the cSH2 and SH3 domains was removed and replaced with a flexible linker; we refer to this internal deletion as Δ25 (*Figure 1—figure supplement 1*, also see Materials and methods). The crystallized construct therefore contains residues 21–765 and 791–1215 of PLC-γ1.

The cSH2/SH3 domain loop contains Tyr783, which is required for phosphorylation-dependent activation of PLC-γ1. However, we have demonstrated previously that this loop is not directly required for autoinhibition (*Gresset et al., 2010*), and consistent with this notion, the crystallized form of PLC-γ1 was autoinhibited in cells (*Figure 1—figure supplement 2*). While nuanced differences in regulation between the wild-type and crystallized version of PLC-γ1 cannot be excluded, the latter faithfully recapitulated the mutational activation of the wild-type enzyme (*Figure 1—figure supplement 2*). In addition, deletion of the cSH2/SH3 domain loop had no measurable effect on the hydrodynamic radius or specific activity of the purified protein (*Figure 1—figure supplement 2*), further demonstrating that removal of the loop did not significantly alter the biochemical properties of the enzyme.

The 2.5 Å crystal structure (*Supplementary file 1*) of essentially full-length rat PLC-γ1 readily explains the autoinhibition of the PLC-γ isozymes. In particular, the regulatory array sits 'atop' the conserved catalytic core and blocks the core from productively engaging membranes (*Figure 1b*). In addition, the overall structure is highly electronegative (*Figure 1—figure supplement 3*), and this property will also inhibit lipase activity by disfavoring interactions with negatively charged membranes. In particular, the overall negative charge of the PH domain indicates that it is unlikely to bind phosphatidylinositol 3,4,5-trisphosphate as previously reported (*Falasca et al., 1998*). For PLCs to hydrolyze membrane-embedded $PIP_2$, the hydrophobic ridge of the catalytic TIM barrel must insert into lipid bilayers (*Ellis et al., 1998*). However, in the case of PLC-γ1, the hydrophobic ridge interacts with portions of the sPH domain in the regulatory array; this arrangement is expected to effectively block membrane engagement. The active site sits beneath the hydrophobic ridge and is readily located by the bound $Ca^{2+}$ cofactor (*Figure 1—figure supplement 2*). As implied by the visibility of the $Ca^{2+}$ cofactor, the active site is fully solvent-exposed and could presumably hydrolyze soluble substrates not embedded in lipid bilayers.

Two major interfaces lock the regulatory array on top of the catalytic core. The first is the aforementioned sPH domain interacting with the hydrophobic ridge of the TIM barrel (*Figure 1c*). Here, residues from the sPH domain interdigitate with residues of the hydrophobic ridge in a 'zipper-like' arrangement. A second interface is formed between loops of the cSH2 domain and the C2 domain of the catalytic core (*Figure 1d*). In this case, the BG and EF loops of the cSH2 domain clasp prominent turns of the C2 domain—almost as if the loops are pinching the C2 domain. The pinched region of the C2 domain is an additional membrane anchor point in the PLC-δ isozymes, where $Ca^{2+}$ mediates between the C2 domain and negatively charged membranes (*Ananthanarayanan et al., 2002*; *Lomasney et al., 2012*). Based on sequence conservation and overall charge distribution, this region of the C2 domain of PLC-γ1 also seems likely to interact with $Ca^{2+}$ and membranes, although this idea has not been tested. The analogous portion of the C2 domain of PLC-γ2 is anticipated to engage $Ca^{2+}$ in a similar manner. Of note, this loop was implicated in the $Ca^{2+}$-dependent translocation of PLC-γ2 to the plasma membrane necessary for amplification of the $Ca^{2+}$ signaling cascade in B cells (*Nishida et al., 2003*). The two interfaces between the regulatory array and catalytic core do not overlap, but the sPH and cSH2 domains within the regulatory array brace each other through the C-terminal portion of the SH3 domain that lies between them (*Figure 1e*): picture an arch with the tail end of the SH3 domain acting as the keystone.

The structure of full-length and autoinhibited PLC-γ1 immediately evokes a straightforward mechanism for its activation upon tyrosine phosphorylation. The same BG and EF loops of the cSH2 domain that pinch the C2 domain are also used to engage pTyr783 and surrounding regions (*Hajicek et al., 2013*) (*Figure 1f*, *Figure 1—figure supplement 4*). Therefore, when Tyr783 is phosphorylated, we propose that it will compete with the C2 domain for binding to the cSH2 domain. This competition would presumably disengage the cSH2 domain from the C2 domain to initiate a rearrangement of the regulatory array relative to the catalytic core. Moreover, perturbations at the interface of the cSH2 and C2 domains may propagate to the interface between the sPH domain and

TIM barrel through keystone residues of the SH3 domain tail to amplify the original structural rearrangements. Indeed, the structure suggests that a wholesale rearrangement of the regulatory array relative to the catalytic core is required for productive membrane engagement by PLC-γ1 (*Figure 1g*). This idea is consistent with previous studies using small-angle X-ray scattering (SAXS) that showed phosphorylation of PLC-γ1 is coupled to a conformational change that has yet to be defined (*Bunney et al., 2012*).

The SAXS studies were also used by Bunney and colleagues to analyze the arrangement of domains within PLC-γ1. They posited a fundamentally distinct arrangement of domains relative to the crystal structure presented here. In particular, the sPH and cSH2 domains were modeled as occupying the central volume of the SAXS envelope with the nSH2 and SH3 domains assuming flanking positions. In this model, the sPH domain does not contact the PLC core. This arrangement of domains would necessarily require a different mode of autoinhibition with the main autoinhibitory interface formed between the cSH2 domain and the TIM barrel.

The overall structure of PLC-γ1 also supports the multivalent scaffolding properties of the PLC-γ isozymes required for proper signaling (*Lemmon and Schlessinger, 2010*; *Rouquette-Jazdanian et al., 2012*; *Timsah et al., 2014*). In particular, both the nSH2 and SH3 domains are organized within the overall structure for unfettered access to cognate ligands (*Figure 2*, *Figure 1—figure supplement 4*). For example, the phosphotyrosine-binding pocket of the nSH2 domain is fully solvent exposed and the major site for engagement of phosphorylated RTKs (*Bae et al., 2009*; *DeBell et al., 1999*; *Poulin et al., 2000*). Likewise, the canonical polyproline-binding site of the SH3 domain is positioned to readily engage various proteins. Relevant examples include the scaffolding protein SLP-76 (*Deng et al., 2005*), the E3 ubiquitin ligase Cbl (*Tvorogov and Carpenter, 2002*), and the guanine nucleotide exchange factor Vav1 (*Braiman et al., 2006*; *Knyazhitsky et al., 2012*; *Sherman et al., 2016*)—all of which must be engaged by PLC-γ1 for the proper clustering of T cell receptors and subsequent downstream signaling. PLC-γ2 mediates similar clustering in response to active B cell receptors (*Wang et al., 2014*) and presumably will be structurally similar to PLC-γ1.

In addition, the monomeric GTPase Rac2 binds the sPH domain of PLC-γ2 to elevate lipase activity (*Piechulek et al., 2005*; *Walliser et al., 2008*) and the equivalent surface within the sPH domain of PLC-γ1 is fully exposed to solvent (*Figure 1—figure supplement 4*). This last observation suggests that binding of Rac2 would not disrupt the overall structure of autoinhibited PLC-γ2. Rather, Rac2 may stabilize an active conformation of PLC-γ2 once PLC-γ2 is engaged with membranes as previously suggested (*Walliser et al., 2016*).

In counterpoint to the above examples, the canonical phosphotyrosine-binding site of the cSH2 domain is buried in the structure of PLC-γ1. This site is presumably reserved as the 'trigger' for phospholipase activation upon engagement of pTyr783. Therefore, pTyr783 is suggested to function as an intramolecular ligand that can effectively compete for the buried surface of the cSH2 domain. This situation is in contrast to intermolecular competitors such as kinases that would need to overcome substantial entropic penalties in order to bind the cSH2 domain.

Tyr783 in PLC-γ1 is presumed to be the primary site of phosphorylation coupled to enzyme activation (*Gresset et al., 2010*; *Kim et al., 1991*). Eight additional tyrosines are phosphorylated (positions 186, 472, 481, 771, 775, 959, 977, and 1254), but these sites appear dispensable for RTK-dependent activation in cells (*Bunney et al., 2012*). In contrast, activation of PLC-γ1 by soluble tyrosine kinases requires phosphorylation of both Tyr775 and Tyr783 (*Serrano et al., 2005*) and this situation is similar for PLC-γ2 where the analogous tyrosines (positions 753, 759) are also phosphorylated during phospholipase activation (*Humphries et al., 2004*; *Ozdener et al., 2002*; *Rodriguez et al., 2001*; *Watanabe et al., 2001*). How dual sites of phosphorylation cooperate to drive phospholipase activity is an open question but presumably shares aspects of regulation described above. Additional tyrosines (positions 1197, 1217) in PLC-γ2 are also phosphorylated and implicated in regulation (*Watanabe et al., 2001*), but these sites are not conserved in PLC-γ1.

## Interfacial regulation

While the structure of PLC-γ1 strongly suggests that it must undergo a substantial rearrangement in order to gain access to its membrane-resident substrate, PIP$_2$, this idea is speculative without substantiation. We formally tested this idea using two bespoke fluorescent substrates of mammalian PLCs (*Figure 3*). The first case, WH-15, is a soluble analogue of PIP$_2$ (*Huang et al., 2011*). It is predicted to have unimpeded access to the active site of PLC-γ1 and mutations assumed to relieve

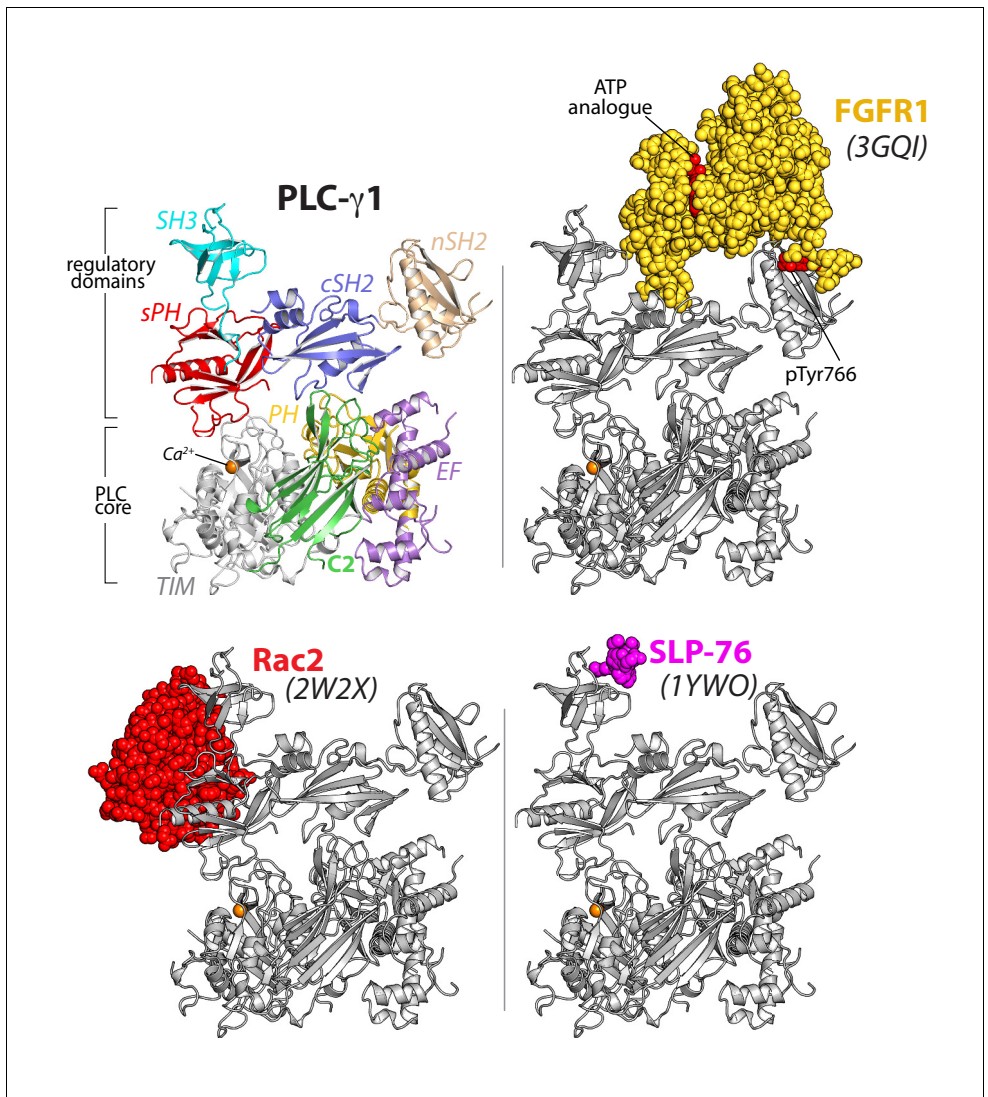

**Figure 2.** The regulatory domains of PLC-γ1 are organized to integrate multiple inputs simultaneously. Structures of fragments of PLC-γ1 or -γ2 bound to biologically relevant proteins were superimposed on the structure of PLC-γ1 (gray ribbon) and partner proteins highlighted in color as space filling models. PDB accession numbers are listed in parentheses and the domain architecture of PLC-γ1 is shown at upper left in the same orientation.

autoinhibition by wholesale rearrangement should not affect basal specific activity for the hydrolysis of WH-15. This is in fact the case since wild-type PLC-γ1 and a set of mutated forms that are constitutively active in cells (*Hajicek et al., 2013*) (also see below) have essentially identical capacity to hydrolyze WH-15 in vitro (*Figure 3a*). In contrast, XY-69 is a fluorescent substrate of PLCs that was specifically designed to embed into lipid bilayers (*Huang et al., 2018*). When XY-69 in lipid vesicles was presented to the same set of PLCs, there was now a dramatic difference in hydrolytic rates (*Figure 3b*). Wild-type PLC-γ1 had very low specific activity, while the mutated forms were up to 30-fold more active. This discrimination presumably reflects the capacity of mutations to disrupt the interface between the regulatory domains and the catalytic core to favor a form of PLC-γ1 better able to engage PIP$_2$ in membranes. Discrimination was greatly diminished—albeit not completely eliminated—when XY-69 was solubilized in detergent micelles (*Figure 3c*). These results are consistent with the postulation that autoinhibition arises from the overall spatial arrangement of PLC-γ1 that prevents it from productively engaging membranes. Mutations that destabilize this arrangement are proposed to concomitantly relieve autoinhibition and allow PLC-γ1 better access to membranes and PIP$_2$.

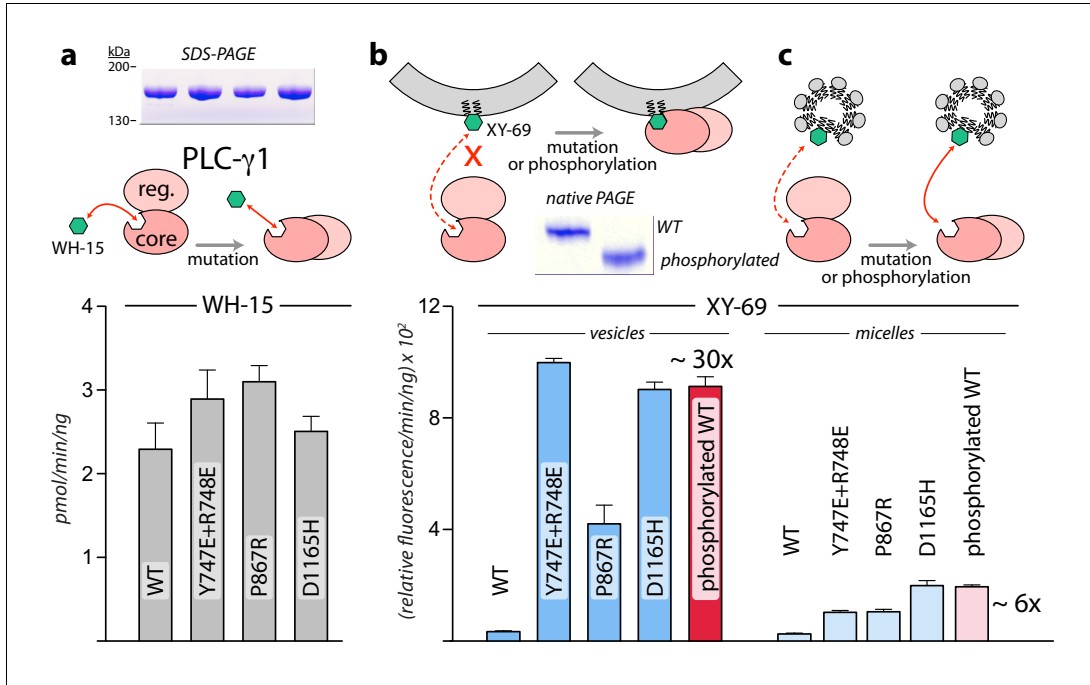

**Figure 3.** Interfacial regulation of purified forms of PLC-γ1. (**a**) Specific activities measured with the soluble substrate, WH-15 (3 μM). Purified PLC-γ1 variants (2 μg) shown in *inset* in the same order as bar chart. Data represent the mean ± SEM calculated from three independent experiments. (**b–c**) Quantification of phospholipase activity at lipid interfaces. The membrane-associated substrate XY-69 (5 μM) was incorporated into phospholipid vesicles containing 220 μM PE and 20 μM PIP$_2$ (*vesicles*) or detergent-mixed micelles containing 0.5% w/v sodium cholate (*micelles*) prior to the addition of indicated forms of PLC-γ1 (wild-type PLC-γ1, 1 nM; mutant forms, 0.5 nM). Phospholipase activity determined by quantifying XY-69 hydrolysis in real-time and presented as the mean ± SEM of three independent experiments. Phosphorylation of PLC-γ1 by a constitutively active version of the FGFR2 kinase domain confirmed by native PAGE followed by Coomassie Brilliant blue staining (*inset*).

Accelerated molecular dynamics (aMD) simulations support the proposed mechanism of activation. In particular, all-atom simulations of PLC-γ1 reproducibly highlighted a flexible set of regulatory domains relative to a virtually static catalytic core (*Figure 4a*, *Figure 4—figure supplement 1*). Moreover, this flexibility increased for simulations of a constitutively active mutant form of PLC-γ1 harboring a single substitution (D1165H) within the C2 domain at the interface with the phosphotyrosine-binding site of the cSH2 domain (*Figure 4—figure supplement 2*). Of note, D1165H corresponds to the D1140G substitution in PLC-γ2; PLC-γ2(D1140G) has been identified in patients with relapsed chronic lymphocytic leukemia treated with ibrutinib (*Burger et al., 2016*; *Landau et al., 2017*). For both wild-type and mutant PLC-γ1, the correlated motions indicate that the regulatory domains tended to move as a relatively rigid block (*Figure 4b*).

Comparisons of average structures derived from the aMD simulations highlight increased disorganization within the interface between the cSH2 and C2 domains upon mutation (*Figure 4c*). For example, Asp1165 resides within the β5/β6 turn of the C2 domain where it participates in two hydrogen bonds that stabilize the turn that forms a major part of the interface with the cSH2 domain. Substitution of Asp1165 to His (D1165H) disrupts the proximal hydrogen-bonding network and results in the partial unfolding of the β5 and β6 strands of the C2 domain during simulations. The collapse of this region is linked to an approximately 30° rotation of the cSH2 domain as it moves toward the C2 domain by approximately 10 Å. The relative movements of the C2 and cSH2 domains are propagated to the rest of the regulatory array due to its propensity to move as a block. Movements are essentially identical for a constitutively active mutant form of PLC-γ1 harboring two substitutions (Y747E+R748E) within the phosphotyrosine-binding site of the cSH2 domain and on the opposite side of the interface from Asp1165 (*Figure 4—figure supplements 1*, *2* and *3*). This result suggests that diverse mutations within the cSH2/C2 domain interface will favor similar movements.

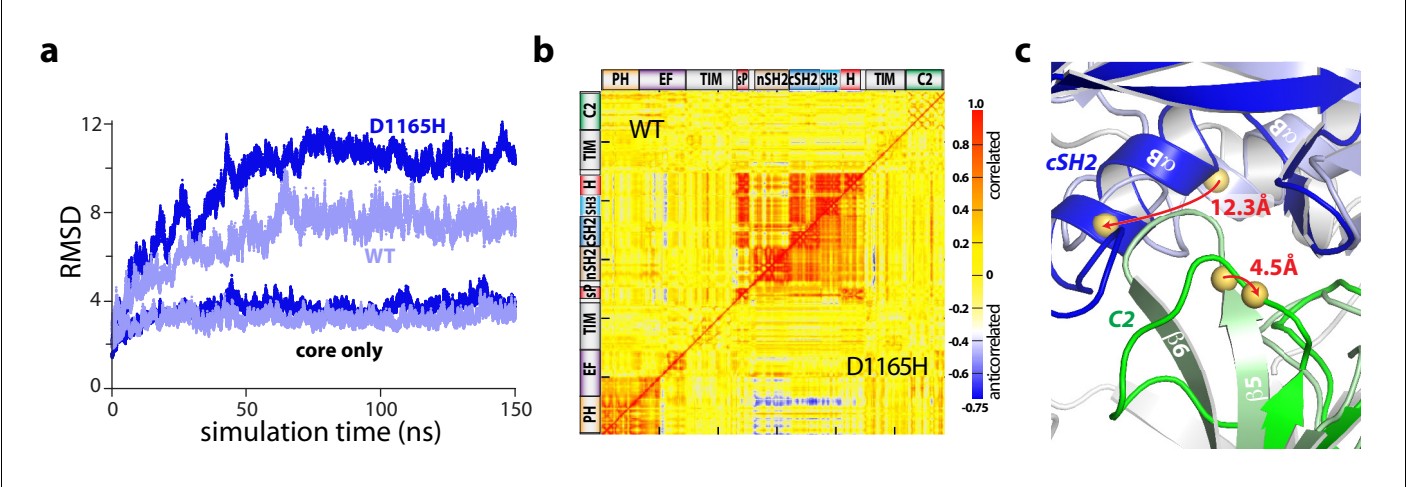

**Figure 4.** The regulatory domains of PLC-γ1 are dynamic in aMD simulations. (a) Root mean square deviations (RMSD) of backbone atoms for the indicated trajectories relative to the starting model of autoinhibited PLC-γ1. For comparison, the equivalent RMSDs for the PLC core ('core only') are also shown. (b) Correlation matrix for pairs of residues in PLC-γ1 and PLC-γ1(D1165H). Correlated motions were calculated over the first 75 nanoseconds of each simulation. (c) Superimposition of the average structures of PLC-γ1 and PLC-γ1(D1165H). Structures were calculated over 75–150 nanoseconds of each simulation. Domains in wild-type PLC-γ1 and PLC-γ1(D1165H) are shown in light and dark colors, respectively; the remainder of the proteins are gray. Red arrows indicate displacement of select Cα atoms (yellow spheres).

The online version of this article includes the following figure supplement(s) for figure 4:

**Figure supplement 1.** Activated forms of PLC-γ1 have highly mobile regulatory domains in aMD simulations.

**Figure supplement 2.** Substitutions of PLC-γ1 analyzed in aMD simulations.

**Figure supplement 3.** Point mutations in the cSH2 domain recapitulate the dynamics of the regulatory domains in aMD simulations.

## PLC-γ isozymes in cancers

The PLC-γ isozymes are frequently mutated in several leukemias (*Burger et al., 2016*; *Kataoka et al., 2015*) and lymphomas (*Choi et al., 2015*; *da Silva Almeida et al., 2015*; *Kiel et al., 2015*; *Vaqué et al., 2014*). In particular, PLC-γ1 is the most frequently mutated protein in adult T cell leukemia/lymphoma (*Kataoka et al., 2015*). In this disease, sites of substitution in PLC-γ1 are found throughout the entire primary sequence with clusters at several hotspots (*Figure 5a*). This rather uninformative arrangement is dramatically clarified when the entire set of substitutions is mapped onto the structure of autoinhibited PLC-γ1 (*Figure 5b*). Now, the majority of sites localize to the interfaces formed between the PLC core and the regulatory array. This three-dimensional clustering strongly suggests that most cancer-associated substitutions in PLC-γ1 disrupt the placement of the regulatory domains atop the core to disfavor autoinhibition. Indeed, in a panel of PLC-γ1 isozymes expressed in HEK293 cells, cancer-associated substitutions at these interfaces produced a spectrum of constitutively active phospholipases—sometimes exceeding 1500-fold greater activity than wild-type PLC-γ1 (*Figure 5c*). Cancer-associated mutations within the equivalent regions of PLC-γ2 produced similar enhancements, indicating conserved regulation between the two isozymes (*Figure 5—figure supplement 1*).

Cancer-derived mutations outside the autoinhibitory interfaces generally produced the smallest increases in basal lipase activities—but these increases were nonetheless significant in comparison to the wild-type isozyme (*Figure 5c*, inset). How might these additional mutations lead to constitutive phospholipase activity? Based on the sites of mutation within the structure of autoinhibited PLC-γ1, three mechanisms are likely. First, substitutions may increase the affinity of the active form of PLC-γ1 for membranes. This option is likely the case for R48W located in the PH domain near the presumed interface with membranes. A similar mode leading to elevated phospholipase activation was proposed for a substituted form of PLC-γ2 that causes arthritis in mice and has increased affinity for membranes relative to wild-type PLC-γ2 (*Everett et al., 2009*). Second, substitutions might disrupt interactions provided by the keystone residues of the SH3 domain that buttress the organization of the sPH and cSH2 domains needed to maintain autoinhibition. Representative substitutions include

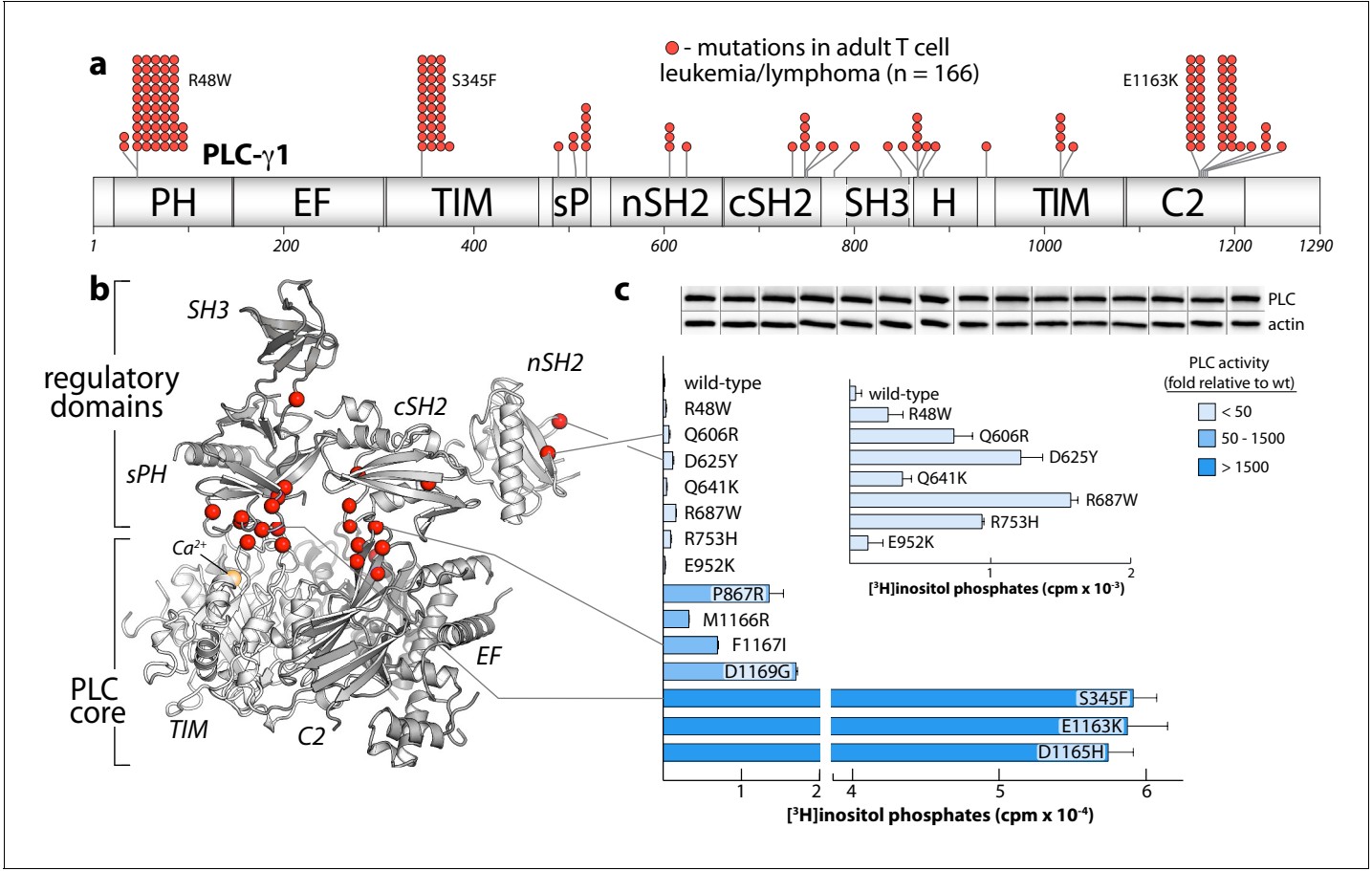

**Figure 5.** Substitutions of PLC-γ1 found in cancers activate the enzyme. (a) Position (n = 26) and frequency of substitutions (red spheres) in PLC-γ1 for a cohort of 370 patients with adult T cell leukemia/lymphoma. (b) Mutations from (a) mapped onto the structure of PLC-γ1. (c) Basal phospholipase activity of mutant forms of PLC-γ1 in cells. Data represent the mean ± SEM of triplicate samples from a single experiment representative of three independent experiments. *Inset* shows mutant forms of PLC-γ1 with the lowest relative basal activity. Immunoblots of cell lysates are presented in the same order as the bar graph.

The online version of this article includes the following figure supplement(s) for figure 5:

**Figure supplement 1.** PLC-γ2 is constitutively activated by substitutions found in cancers.

**Figure supplement 2.** Cancer-associated substitutions within the 'keystone' residues of the SH3 domain activate PLC-γ1.

R687W and R753H and additional examples are found in both PLC-γ1 (*Figure 5—figure supplement 2*) and -γ2 (*Figure 5—figure supplement 1*). Of note, R687W is analogous to R665W in PLC-γ 2 and arises in patients with relapsed chronic lymphocytic leukemia treated with ibrutinib (*Woyach et al., 2014*). Finally, mutations within the nSH2 domain, for example Q606R and D625Y, are near the binding site for phosphotyrosine (*Bae et al., 2009*) and may increase affinity for phosphorylated kinases.

The PLC-γ isozymes are normally activated upon phosphorylation, especially by diverse growth factor receptors. Therefore, the cancer-associated mutations in PLC-γ1 were further tested for effects on lipase activity after co-expression of PLC-γ1 and EGFR (*Figure 6a*). In all cases, a high concentration of EGF used to activate the receptor produced elevated lipase activity relative to wild-type PLC-γ1. This result indicates an untapped reserve of lipase activity that is, at least partially, released by these cancer-associated mutations in response to EGF. This point is further emphasized for lipase responses measured at varying concentrations of EGF for a representative subset of mutant PLC-γ1 isozymes with varying levels of constitutive activation (*Figure 6b*, upper graph). Both P867R and D1165H occur at the autoinhibitory interfaces and produced substantially elevated lipase activity relative to wild-type PLC-γ1 at all concentrations of EGF. In contrast, R48W occurs at the

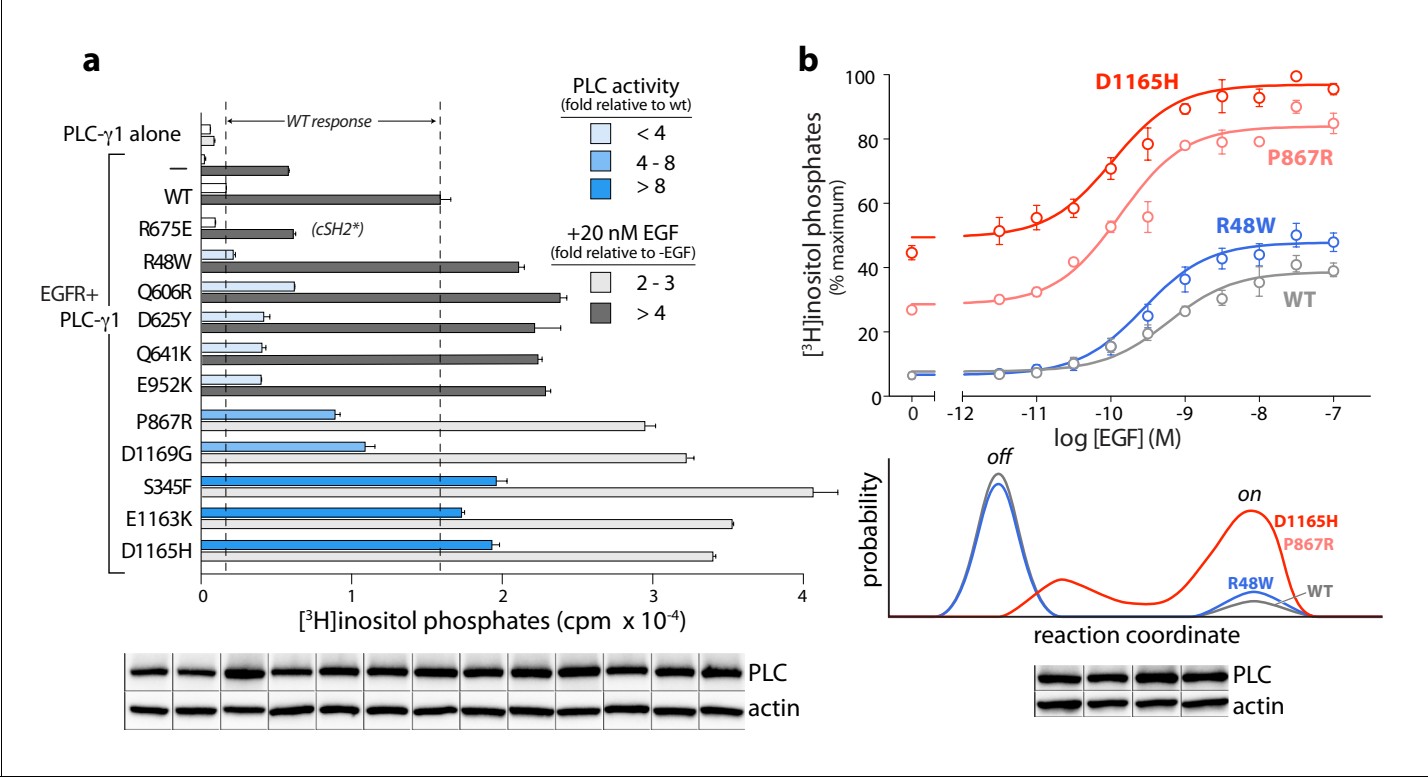

**Figure 6.** Cancer-associated substitutions prime PLC-γ1 for activation by EGFR in cells. (a) Receptor-dependent activation of mutant forms of PLC-γ1. A substitution that abolishes phosphorylation-dependent activation of PLC-γ1 is indicated (cSH2*). Data are the mean ± SEM of triplicate samples from one experiment representative of two independent experiments. (b) EGF concentration-effect curves for wild-type and select mutant forms of PLC-γ1. Data are presented as the mean ± SEM of single data sets pooled from three independent experiments. Hypothetical reaction coordinates for each form of PLC-γ1 are shown below. In both panels, immunoblots of cell lysates transfected with PLC-γ1 are presented in the same order as the bar chart (top to bottom).

predicted interface of the active isozyme with membranes and abnormally elevated lipase activity of PLC-γ1(R48W) manifests only at high concentrations of EGF. This functional difference possibly reflects a mechanistic difference: P867R and D1165H likely destabilize the inactive ensemble of PLC-γ1 while simultaneously favoring active forms of the isozyme; in contrast, R48W has no substantive effect on the inactive population under these conditions and presumably only stabilizes the active isozyme once bound to membranes (*Figure 6b*, lower graph).

Regardless of the mechanistic details, these functional results suggest important biological ramifications. Namely, the lipase activity of mutant PLC-γ isozymes should be dependent on cellular context. For example, PLC-γ1 or -γ2 harboring mutations such as R48W that preferentially stabilize active ensembles may be essentially quiescent until the isozymes are activated by phosphorylation. These situations are relatively nuanced in comparison to more robustly activating mutations, for example P867R and D1165H, that disrupt core aspects of autoinhibition. However, the more subtly activating substitutions may nonetheless contribute to cancer in cells with high levels of active kinases such as upon the overexpression of EGFR or other growth factor receptors—situations with widespread clinical relevance (*Wilson et al., 2012*).

## Discussion

The structure of full-length, autoinhibited PLC-γ1 provides a first clear view of the regulated activation of the PLC-γ isozymes. The overall picture is of a catalytic core that is conserved among all PLCs and that is prevented from spuriously hydrolyzing PIP$_2$ by a set of interdependent regulatory domains stationed to preclude access of the active site to membranes. Additionally, the regulatory

domains are organized to integrate numerous molecular inputs that ultimately control phospholipase activity and mediate necessary scaffolding functions (*Figure 7*). Importantly, the nSH2 domain is optimally positioned to readily bind phosphorylated kinases and align them to promote the phosphorylation of Tyr783 needed for activation of PLC-γ1. Although capable of engaging phosphorylated portions of kinases (*Groesch et al., 2006*), the equivalent surface of the cSH2 domain is buried through interactions with the C2 domain and is unlikely to initiate engagement of kinases as previously suggested (*Huang et al., 2016*).

However, the two SH2 domains might work in concert upon receptor engagement to facilitate the binding of phosphorylated Tyr783 to the cSH2 domain. This idea is supported by the comparison of the full-length structure of PLC-γ1 with a structure of the two SH2 domains of PLC-γ1 bound to the phosphorylated kinase domain of fibroblast growth factor receptor 1 (FGFR1) (*Bae et al., 2009*). Based on this comparison, the βA/αA loop of the nSH2 domain is rearranged to accommodate pTyr766 of FGFR1 and this rearrangement leads to additional movements of the cSH2 domain (*Figure 7—figure supplement 1*). In the full-length structure, equivalent movements upon binding FGFR1 would open the surface of the cSH2 domain that binds pTyr783, effectively priming it to engage pTyr783. Engagement of pTyr783 by the cSH2 domain is presumed to unlatch the cSH2 domain from the catalytic core and initiate what is likely to be a relatively massive rearrangement of the regulatory domains with respect to the core before the core can engage membranes and hydrolyze $PIP_2$.

The model of activation described above provides the mechanistic underpinnings for understanding the mutational landscape of the PLC-γ isozymes associated with disease. In particular, most of the substitutions and small deletions in these isozymes that are linked to cancers (*Burger et al., 2016*; *Kataoka et al., 2015*; *Woyach et al., 2014*) or autoimmune disease (*Ombrello et al., 2012*) occur at the interfaces between the core and regulatory domains based on the structure of autoinhibited PLC-γ1. These mutations disrupt these interfaces, release autoinhibition, and favor conformations that engage membranes to promote constitutive phospholipase activity. Shifted conformational equilibria may also explain the supra-activation of mutant forms of PLC-γ1 by EGFR. That is, mutant forms of PLC-γ1 that are predisposed to be 'open' may also have a greater propensity to bind EGFR and a lower probability of turning off.

Constitutive activation varies greatly, ranging from approximately 10-fold to over 1500-fold, with important cellular implications: highly active forms are expected to drive downstream signaling

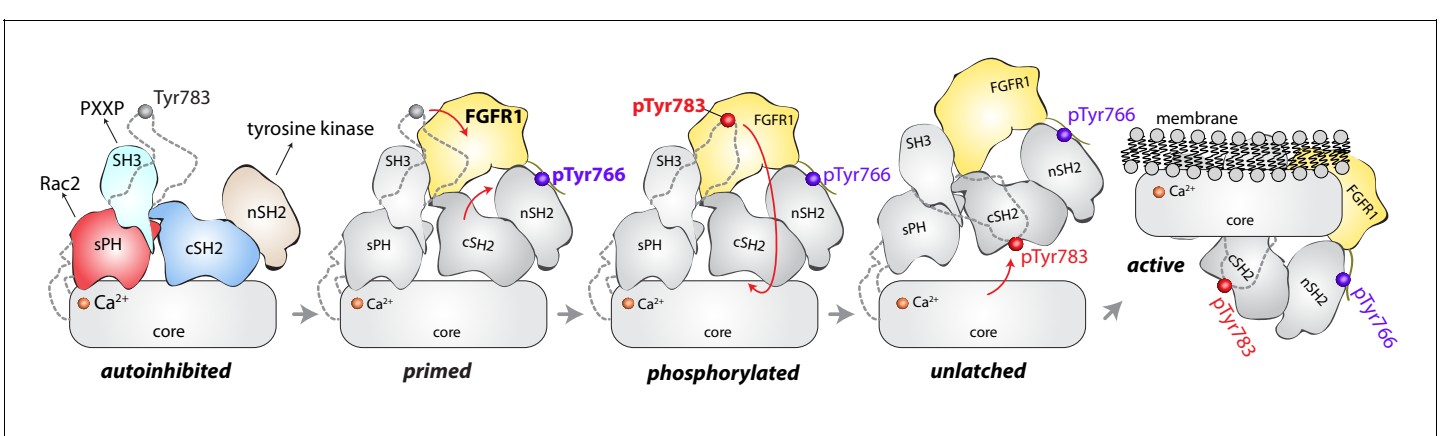

**Figure 7.** Model for phosphorylation-induced activation of PLC-γ1. In the basal state, the nSH2 domain of PLC-γ1 mediates recruitment of the autoinhibited enzyme to activated receptor tyrosine kinases, for example FGFR1. The nSH2 domain binds to phosphorylated Tyr766 (pTyr766) in the C-terminal tail of FGFR1, and its kinase domain acts as a lever to destabilize the interaction between the cSH2 and C2 domains of PLC-γ1, priming the lipase for phosphorylation-dependent activation. PLC-γ1 is subsequently phosphorylated on Tyr783, and the engagement of pTyr783 by the cSH2 domain results in the full dissociation of the cSH2 domain from the C2 domain. Importantly, the phosphorylation of Tyr783 and its subsequent engagement by the cSH2 domain is predicted to induce a large-scale rearrangement of the regulatory domains with respect to the core before the phospholipase can hydrolyze membrane-resident $PIP_2$.

The online version of this article includes the following figure supplement(s) for figure 7:

**Figure supplement 1.** Proposed mechanism for priming of PLC-γ1 by FGFR1.

under all circumstances while more subtly active forms are likely to promote disease only within specific cellular contexts. An excellent example of this latter class includes mutated forms of PLC-γ2 (R665W or L845F) that arise in patients treated with ibrutinib. B cells harboring either mutant of PLC-γ2 possess normal calcium homeostasis until B cell receptors are activated, at which point intracellular calcium levels rise and remain elevated, while in the equivalent wild-type case, calcium homeostasis is rapidly reestablished (*Woyach et al., 2014*). Both mutant forms have essentially wild-type phospholipase activity at low levels of expression but are hypersensitive to activation by Rac2 (*Walliser et al., 2016*). These results suggest that once active, the mutant forms of PLC-γ2 are stabilized at membranes by binding to Rac2. That is, the mutations are cryptic until the isozymes are activated, at which point the mutated PLC-γ2 isozymes are slow to return to their autoinhibited state.

Similar context-dependent activation of mutant forms of the PLC-γ isozymes should also occur upon activation by kinases. PLC-γ1(R48W) provides an example: it has essentially wild-type phospholipase activity until co-expressed with high levels of active EGFR. Analogous scenarios may be widespread in cancers where tyrosine kinases are constitutively active upon substitution, truncation, fusion, or overexpression—all conditions shown to activate PLC-γ1 (*Arteaga et al., 1991*; *Peles et al., 1991*). Alternatively, the activation of RTKs by stromal components contributes to treatment-resistant cancers (*Straussman et al., 2012*) and roles for wild-type and mutant PLC-γ isozymes under these conditions need to be explored.

On a final note, the interfacial regulation of the PLC-γ isozymes suggests promising avenues for their isozyme-specific, allosteric modulation by small molecules to advance related chemical biology and on-going drug discovery. Namely, compounds that inhibit the movement of the regulatory domains relative to the catalytic core should also prevent membrane engagement and consequent $PIP_2$ hydrolysis. Such compounds might treat cancers and immune diseases driven by constitutively active forms of the PLC-γ isozymes. Conversely, allosteric modulators that stabilize active forms of the PLC-γ isozymes might bolster immunotherapies (*Guittard et al., 2018*) or provide promising leads for the treatment of Alzheimer's disease where a naturally occurring hypermorphic variant of PLC-γ2 is linked to protection from this disease (*Magno et al., 2019*).

# Materials and methods

**Key resources table**

| Reagent type (species) or resource | Designation | Source or reference | Identifiers | Additional information |
|---|---|---|---|---|
| Strain, strain background (*Escherichia coli*) | Rosetta2 (DE3) pLysS | Novagen | Cat# 71403 | Chemically competent cells |
| Cell line (*Homo sapiens*) | HEK293 | American Type Culture Collection | Cat# CRL-1573 RRID:CVCL_0045 | |
| Cell line (*Trichoplusia ni*) | HighFive | Invitrogen | Cat# B85502 RRID:CVCL_C190 | |
| Antibody | anti-HA epitope (mouse monoclonal) | BioLegend | Cat# 901513 RRID:AB_2565335 | WB (1:1000) |
| Antibody | anti-β-actin (mouse monoclonal) | SigmaAldrich | Cat# A1978 RRID:AB_476692 | WB (1:4000) |
| Recombinant DNA reagent | pcHALIC-PLC-γ1 (plasmid) | this paper | | Vector is a modified version of pcDNA3.1 containing HA tag |
| Recombinant DNA reagent | pcHALIC-PLC-γ2 (plasmid) | this paper | | Vector is a modified version of pcDNA3.1 containing HA tag |
| Recombinant DNA reagent | p15LIC2-FGFR2K E565A (plasmid) | PMID: 20807769 | | Vector is a modified version of pET15b containing His$_6$ tag and TEV cleavage sequence |
| Recombinant DNA reagent | pFBLIC2-PLC-γ1(21–1215) (plasmid) | this paper | | Vector is a modified version of pFastBacHT1 containing His$_6$ tag and TEV cleavage sequence |

*Continued on next page*

*Continued*

| Reagent type (species) or resource | Designation | Source or reference | Identifiers | Additional information |
|---|---|---|---|---|
| Recombinant DNA reagent | pFBLIC2-PLC-γ1(21–1215) Δ25 (plasmid) | this paper | | Vector is a modified version of pFastBacHT1 containing His$_6$ tag and TEV cleavage sequence |
| Recombinant DNA reagent | pcDNA3-EGFR (plasmid) | this paper | | Original source: Dr. H. Shelton Earp, UNC-Chapel Hill |
| Peptide, recombinant protein | human epidermal growth factor | Invitrogen | Cat# PHG0313 | |
| Software, algorithm | HKL2000 | PMID: 27799103 | | |
| Software, algorithm | Phenix | PMID: 20124702 | RRID:SCR_014224 | Version 1.10.1–2155 |
| Software, algorithm | WinCoot | PMID: 20383002 | RRID:SCR_014222 | Version 0.8.2 |
| Software, algorithm | PyMOL | pymol.org | RRID:SCR_000305 | Version 1.8.2 |
| Software, algorithm | Modeller | PMID: 10940251 | RRID:SCR_008395 | Version 9.16 |
| Software, algorithm | Amber | ambermd.org | RRID:SCR_014230 | Version 14 |
| Other | WH-15 | PMID: 21158426 | | Fluorescent PIP$_2$ analogue, soluble |
| Other | XY-69 | PMID: 29263090 | | Fluorescent PIP$_2$ analogue, membrane-associated |

## DNA constructs

### Mammalian expression constructs

Gibson Assembly cloning (*Gibson et al., 2009*) was used to introduce single amino acid substitutions into full-length rat PLC-γ1 (UniProt accession number P10686; 96% identical to human PLC-γ1) and human PLC-γ2 (P16885) in a modified pcDNA3.1 expression vector that incorporates an HA epitope tag at the N-terminus of the expressed protein. The entire open-reading frame of all constructs was confirmed by automated dideoxy sequencing.

### Bacterial expression constructs

The construct encoding a constitutively active form of the soluble kinase domain of FGFR2 (FGFR2K E565A) was described previously (*Gresset et al., 2010*).

### Baculovirus transfer vectors

PLC-γ1(21–1215) was amplified from full-length rat PLC-γ1 by PCR and then subcloned into a modified pFastBacHT1 vector that incorporates a His$_6$ tag followed by a tobacco etch virus (TEV) protease recognition sequence at the N-terminus of the expressed protein. Transfer vectors encoding PLC-γ1 (21–1215) harboring the P867R or D1165H substitutions were generated similarly, using the full-length mutant forms of PLC-γ1 as PCR templates.

The Δ25 deletion, which replaces residues 766–790 of PLC-γ1 with a Ser-Gly-Ser linker, was introduced into the transfer vector encoding PLC-γ1(21–1215) by standard primer-mediated mutagenesis. PLC-γ1(21–1215) and PLC-γ1(21–1215) Δ25 were amplified by PCR and subcloned into the modified pcDNA3.1 expression vector described above using a ligation-independent cloning strategy (*Stols et al., 2002*).

## Protein expression and purification

### PLC-γ1(21–1215) Δ25

Recombinant baculovirus encoding His$_6$-PLC-γ1(21–1215) Δ25 was prepared using the Bac-to-Bac Baculovirus Expression System according to the manufacturer's protocol (Invitrogen). Four liters of HighFive (*T. ni*) cells at a density of ~$2.0 \times 10^6$ cells/mL were infected with amplified baculovirus stock (10–15 mL/L) and harvested ~60 hr post-infection by centrifugation at 6000 rpm in a Beckman JA-10 rotor at 4°C. All subsequent centrifugation and chromatography steps were performed at 4°C.

The cell pellet was resuspended in 200 mL of ice-cold buffer N1 (20 mM HEPES (pH 7.5), 300 mM NaCl, 10 mM imidazole, 10% v/v glycerol, 0.1 mM EDTA, and 0.1 mM EGTA) supplemented with 10 mM 2-mercaptoethanol and protease inhibitor cocktail prior to lysis using an EmulsiFlex-C5 homogenizer. Crude lysate was centrifuged at 50,000 rpm for 1 hr in a Beckman Ti70 rotor. The supernatant was filtered through a 0.45 μm PES low protein-binding filter and loaded onto a 5 mL HisTrap HP IMAC column equilibrated in buffer N1 supplemented with 5 mM 2-mercaptoethanol. The column was washed with 15 column volumes (CV) of buffer N1, followed by 15 CV of 2.5% buffer N2 (buffer N1 + 1 M imidazole and 5 mM 2-mercaptoethanol). Bound proteins were eluted with 40% buffer N2. Fractions containing PLC-γ1 were pooled and dialyzed overnight in the presence of 2% w/w TEV protease to remove the His$_6$ tag in buffer containing 20 mM HEPES (pH 7.5), 300 mM NaCl, 10% v/v glycerol, 1 mM DTT, 1 mM EDTA, and 0.1 mM EGTA. The sample was subsequently diluted two-fold with buffer N1 and applied to a 5 mL HisTrap HP column. Flow-through fractions containing cleaved PLC-γ1 were pooled, diluted three-fold with buffer Q1 (20 mM HEPES (pH 7.5) and 2 mM DTT), and loaded onto an 8 mL SourceQ anion exchange column equilibrated in 10% buffer Q2 (buffer Q1 + 1 M NaCl). Bound proteins were eluted in a linear gradient of 10–60% buffer Q2 over 50 CV. Fractions containing PLC-γ1 were pooled, concentrated using a VivaSpin 50K MWCO centrifugal concentrator, and applied to a 16 mm x 700 mm HiLoad Superdex 200 size exclusion column equilibrated in buffer containing 20 mM HEPES (pH 7.5), 150 mM NaCl, and 2 mM DTT. Pure PLC-γ1 was concentrated as described above to a final concentration of 40–50 mg/mL, aliquoted, snap-frozen in liquid nitrogen, and stored at −80°C until use.

Wild-type PLC-γ1(21–1215), PLC-γ1(21–1215) P867R, and PLC-γ1(21–1215) D1165H used for biochemical assays were purified using the method described above except that the buffer for size exclusion chromatography was supplemented with 5% v/v glycerol.

### FGFR2K E565A

The soluble kinase domain of FGFR2 (residues 458–778) harboring a His$_6$ tag at its N-terminus was expressed in the Rosetta2 pLysS strain of *E. coli* (Novagen). Cells were grown at 37°C in TB medium containing 0.1 mg/mL ampicillin and 0.034 mg/mL chloramphenicol to an OD$_{600}$ of ~3.0. Protein expression was induced for 2 hr at 30°C with 0.1 mM IPTG (final concentration). Cells were collected by centrifugation, resuspended in lysis buffer (20 mM HEPES (pH 7.5), 300 mM NaCl, 10 mM 2-mercaptoethanol, 10 mM imidazole, 10 mM MgCl$_2$, 10 μM ATP, 10% v/v glycerol, and protease inhibitor cocktail), and lysed using an EmulsiFlex-C5 homogenizer. CHAPS was then added to a final concentration of 0.5% w/v and the lysate incubated at 4°C for 30 min. Soluble lysate was prepared by ultracentrifugation and the kinase domain isolated by IMAC on a HisTrap HP column. The protein was further purified by size exclusion chromatography on a Sephacryl 200 size exclusion column equilibrated in 20 mM HEPES (pH 7.5), 200 mM NaCl, 2 mM DTT, and 5% v/v glycerol. Protein was aliquoted, snap-frozen in liquid nitrogen, and stored at −80°C until use.

## Size-exclusion chromatography coupled to multi-angle light scattering

Multi-angle light scattering measurements were performed using Wyatt DAWN HELEOS II light scattering instrumentation (with Wyatt Optilab T-rEX refractometer and Wyatt dynamic light scattering module) coupled to a Superdex 200 10 mm x 300 mm GL size exclusion column. Following equilibration with buffer containing 20 mM HEPES (pH 7.4), 150 mM NaCl, and 0.02% w/v NaN$_3$, 50 μL of PLC-γ1(21–1215) proteins at 2 mg/mL were loaded onto the column. Data analysis was performed with ASTRA software version 6 (Wyatt Technologies).

## Crystallization of PLC-γ1(21–1215) Δ25

### Native PLC-γ1(21–1215) Δ25

Crystals of PLC-γ1(21–1215) Δ25 were grown initially by sitting drop vapor diffusion. PLC-γ1(21–1215) Δ25 was diluted to 20 mg/mL in buffer containing 20 mM HEPES (pH 7.5), 150 mM NaCl, 5 mM DTT, and 0.25% w/v CHAPSO. Two hundred nanoliters of this protein solution was mixed with 100 nanoliters of reservoir solution (200 mM di-sodium tartrate and 20% w/v PEG 3,350) and equilibrated against a 30 μL reservoir. Crystals grew as a cluster of thin plates and appeared after 9 days at 20°C. Diffraction quality crystals of PLC-γ1(21–1215) Δ25 were grown at 20°C by microseeding hanging drops. Protein solution was prepared by diluting PLC-γ1(21–1215) Δ25 to 40 mg/mL in

buffer containing 20 mM HEPES (pH 7.5), 150 mM NaCl, 5 mM DTT, and 0.25% w/v CHAPSO. Solutions of seed crystals were obtained by vortexing crystals of PLC-γ1(21–1215) Δ25 with a glass bead in buffer containing 200 mM di-sodium tartrate, 25% w/v PEG 3,350, 150 mM NaCl, 5 mM DTT, and 0.25% w/v CHAPSO. Seed crystals were diluted 100-fold in the same buffer prior to use. Drops were prepared by mixing, in order, 1 μL of reservoir solution (12.5% w/v PEG 3,350, 50 mM di-sodium tartrate, and 5% v/v glycerol), 2 μL of protein solution, and 0.5 μL of seed solution. Drops were equilibrated against a 500 μL reservoir. Crystals ~100 μm on the longest edge appeared after 1–2 days and were flash-frozen in liquid nitrogen on nylon loops.

## Gadolinium-derivatized PLC-γ1(21–1215) Δ25

PLC-γ1(21–1215) Δ25 (335 μM) was treated with a 50-fold molar excess of EGTA for 1 hr at 4˚C. The protein was then exchanged into crystallization buffer (20 mM HEPES (pH 7.5), 150 mM NaCl, 5 mM DTT, 5 μM EGTA, 1 mM GdCl$_3$, and 0.25% w/v CHAPSO) using a 7K MWCO Zeba spin desalting column (Thermo Scientific). The final protein concentration in this solution was 36 mg/mL. A solution of seed crystals was prepared as described above except that the buffer contained 50 mM di-sodium tartrate, 20% w/v PEG 3,350, 5% v/v glycerol, 150 mM NaCl, 5 mM DTT, 0.25% w/v CHAPSO, 5 μM EGTA, and 1 mM GdCl$_3$. Drops were prepared by mixing, in order, 1 μL of reservoir solution (12.5% w/v PEG 3,350, 25 mM di-sodium tartrate, and 10% v/v glycerol), 2 μL of protein solution, and 0.5 μL of seed solution. Drops were equilibrated against a 500 μL reservoir. Crystals grew at 20˚C and were transferred from the mother liquor and soaked in buffer containing 25 mM HEPES (pH 7.5), 150 mM NaCl, 5 mM DTT, 0.25% w/v CHAPSO, 12.5% w/v PEG 3,350, 10% v/v glycerol, and 5 mM GdCl$_3$ for 3 min at room temperature. Crystals were mounted on nylon loops and flash-frozen in liquid nitrogen.

## X-ray diffraction data collection and structure determination

X-ray diffraction data were collected on crystals of native PLC-γ1(21–1215) Δ25 at the Southeast Regional Collaborative Access Team (SER-CAT) beamline 22-BM at the Advanced Photon Source at Argonne National Laboratory. One scan totaling 200˚ of data was collected at 100 K on a MAR 200 CCD detector. Each frame was exposed for 10 s and consisted of a 1˚ oscillation.

Gadolinium-derivatized crystals of PLC-γ1(21–1215) Δ25 were used to collect a single-wavelength anomalous diffraction (SAD) dataset at the gadolinium absorption peak (λ = 7,244.3 eV). Data were acquired at SER-CAT beamline 22-ID on a Rayonix MX300-HS CCD detector. Each frame consisted of a 1˚ oscillation and was exposed for 1 s. A total of 240˚ of data were collected at 100 K in 30˚ wedges using an inverse beam strategy. Both sets of diffraction data were indexed, integrated, and scaled using HKL2000 (*Otwinowski and Minor, 1997*).

Phases for the gadolinium-bound form of PLC-γ1(21–1215) Δ25 were solved by SAD using the AutoSol routine in the Phenix software suite (*Adams et al., 2010*). A partial model of this structure was built using AutoBuild (*Terwilliger et al., 2008*) and used as a molecular replacement search model to solve phases for the structure of native PLC-γ1(21–1215) Δ25. The remainder of the model was then built in an iterative process that consisted of manual model building in Coot (*Emsley et al., 2010*) followed by restrained refinement in Phenix. The structure was validated using MolProbity (*Chen et al., 2010*) and molecular representations produced with PyMOL (*Schrodinger LCC, 2019*). Complete data collection and refinement statistics are shown in *Supplementary file 1*.

## Quantification of phospholipase activity in cells

To quantify basal phospholipase activity, HEK293 cells were plated at a density of ~75,000 cells/well in 12-well cluster plates and transiently transfected with 100 ng of vector encoding wild-type or mutant forms of PLC-γ1. Twenty-four hours post-transfection, cells were metabolically labeled overnight in serum-free, inositol-free medium containing 1 μCi of [3H]*myo*-inositol and 10 mM LiCl. Accumulation of [3H]inositol phosphates was quantified as described previously (*Waldo et al., 2010*). In all experiments, counts that accumulated in cells transfected with empty vector (~500–1000 cpm) were subtracted as background.

EGFR-dependent activation of PLC-γ1 and PLC-γ2 was quantified in HEK293 cells transiently co-transfected with 200 ng of vector expressing various forms of the PLC-γ isozymes and 100 ng of vector expressing wild-type EGFR. Cells were metabolically labeled as described above, except LiCl

was omitted from the radiolabeling medium. Cells received a 30 min challenge with the indicated concentrations of recombinant human EGF (Invitrogen) diluted in Hank's Balanced Salt Solution containing 20 mM HEPES (pH 7.5), 10 mM LiCl, and 200 µg/mL fatty acid-free bovine serum albumin (BSA).

Expression of each form of PLC-γ1 and PLC-γ2 was confirmed by immunoblotting of cell lysates using a monoclonal antibody against the HA epitope (BioLegend, clone 16B12). Lysates were also probed with a monoclonal antibody against β-actin (SigmaAldrich, clone AC-15) as a loading control. All immunoblots represent a single exposure from one experiment, and the HA epitope and β-actin were detected on the same blot. Immunoblots were loaded with all mutant versions of PLC-γ1 or PLC-γ2 in numerical order; bands were subsequently cropped and then reordered in Photoshop to reflect the order in which data are presented in bar graphs and dose-response curves. The identity of the HEK293 cell line was not authenticated, and testing for mycoplasma contamination was not performed.

## In vitro quantification of phospholipase activity

### WH-15 fluorogenic assay

Assays utilizing WH-15 as enzyme substrate were performed as described previously (*Charpentier et al., 2014*) with the following modifications. WH-15 (3 µM, final concentration) was solubilized in a final assay buffer containing 50 mM HEPES (pH 7.4), 70 mM KCl, 3 mM EGTA, 2.9 mM CaCl$_2$, 50 µg/mL fatty acid-free BSA, 2 mM DTT, and 0.25% w/v sodium cholate. Baseline fluorescence was stabilized for 10 min, and fluorescence intensity then was quantified for an additional 15 min following addition of various forms of purified PLC-γ1(21–1215) (1 nM, final concentration). Fluorescence intensity was converted to pmol of 6-aminoquinoline using a standard curve, and initial rates of WH-15 hydrolysis were calculated from the slope of linear data points.

### XY-69 fluorogenic assay

All assays with XY-69 (*Huang et al., 2018*) were performed at 30°C in 384-well plates in a PHERAstar multi-mode plate reader. Data were recorded for 30 min at intervals of 1 min using excitation and emission wavelengths of 485 nm and 520 nm, respectively. Fluorescence intensity was normalized to a blank reaction lacking phospholipase, and initial rates of XY-69 hydrolysis were calculated from the slope of the linear portion of the curve. The amount of wild-type and mutant forms of PLC-γ1(21–1215) used in all experiments was adjusted to maintain assay linearity with respect to time and protein concentration.

To prepare mixed micelles, XY-69 (5 µM, final concentration) was dried under a stream of nitrogen and solubilized by sonication in a final assay buffer containing 30 mM HEPES (pH 7.4), 70 mM KCl, 3 mM EGTA, 2.35 mM CaCl$_2$, 2 mM DTT, and 0.5% w/v sodium cholate. PLC-γ1(21–1215) proteins (0.5–1 nM, final concentration) were diluted in 20 mM HEPES (pH 7.4), 50 mM NaCl, 1 mg/mL fatty acid-free BSA, and 2 mM DTT. Reactions were initiated by adding 10 µL of detergent micelles to 2 µL of PLC-γ1.

Phospholipid vesicles were prepared by combining XY-69, porcine brain phosphatidylinositol 4,5-bisphosphate (PIP$_2$), and bovine liver phosphatidylethanolamine (PE) and drying the mixture under a stream of nitrogen. Lipids were resuspended by sonication in 20 mM HEPES (pH 7.4). PLC-γ1(21–1215) proteins (0.5–1 nM, final concentration) were diluted as described above for mixed micelle assays. Assays were initiated by adding 10 µL of phospholipid vesicles to 2 µL of PLC-γ1 and performed in a final assay buffer consisting of 20 mM HEPES (pH 7.4), 70 mM KCl, 3 mM EGTA, 2.35 mM CaCl$_2$, and 2 mM DTT. Final concentrations of XY-69, PIP$_2$, and PE were 5 µM, 20 µM, and 220 µM, respectively.

### In vitro kinase assay

Equimolar concentrations (35 µM) of PLC-γ1(21–1215) and FGFR2K E565A were incubated on ice in buffer containing 20 mM HEPES (pH 7.4), 50 mM NaCl, 10 mM MgCl$_2$, 0.2 mM Na$_3$VO$_4$, 50 ng/mL fatty acid-free BSA, 2 mM DTT, and 0.5 mM ATP. After 1 hr, a portion of the reaction mixture was diluted with 20 mM HEPES (pH 7.4), 50 mM NaCl, 1 mg/mL fatty acid-free BSA, and 2 mM DTT. Phospholipase activity was quantified using XY-69 incorporated into mixed micelles or phospholipid vesicles as described above. The concentrations of PLC-γ1(21–1215) and FGFR2K E565A were both

1 nM in the final reaction mixture. Phosphorylation of PLC-γ1(21–1215) was analyzed by native PAGE on PhastGel homogeneous medium containing 7.5% polyacrylamide followed by staining with Coomassie Brilliant blue.

## [$^3$H]PIP$_2$ hydrolysis assay

Quantification of lipase activity using phospholipid vesicles consisting of 200 μM PE, 20 μM PIP$_2$, and ~5000 cpm/assay [$^3$H]PIP$_2$ was performed as described previously (*Waldo et al., 2010*).

## Homology modeling

### PLC-γ2(14–1190)

A model of PLC-γ1(21–1215) containing residues 766–790 was generated with Modeller v9.16 (*Martí-Renom et al., 2000*) using the structure of PLC-γ1(21–1215) Δ25 as the template. This model of PLC-γ1 was then used as the template to build a model of PLC-γ2(14–1190).

## Molecular dynamics simulations

### Structural model of PLC-γ1(21–1215) Δ25

The X-ray crystal structure of autoinhibited PLC-γ1 included a number of missing regions presumably due to local disorder. Missing regions that were expected to contain secondary structural elements included: i) helix E of EF hand 2 (residues 190–206) and ii) helix E of EF hand 3 (residues 226–246). In order to build the missing helices, HHpred (*Zimmermann et al., 2018*) was used to search for suitable templates. The fragment from the structure of cuttlefish PLC21 (PDB code: 3QR0) (*Lyon et al., 2011*) containing helix F of EF hand 2 through helix F of EF hand 3 provided the best superimposition on the PLC-γ1 structure and was used as a template for building helix E of EF hand 3. Residues 226–233 were deleted from the X-ray structure prior to building a model of this EF hand. The apo structure of troponin C (PDB code: 1TNP) (*Gagné et al., 1995*) containing helix F of EF hand 1 through helix F of EF hand 2 provided the best superimposition on the structure of PLC-γ1 and was used as the template for building helix E of EF hand 2. The remaining loops missing from the structure of PLC-γ1 were built as random coils with no regular secondary structural elements. A structural model of PLC-γ1 containing all residues from Glu21 - Lys1215, except for the shortened activation loop, was generated with Modeller v9.16 using the autoinhibited PLC-γ1 structure as the template. The wild-type PLC-γ1 model was subsequently mutated in PyMOL to generate the PLC-γ1(D1165H) model used for molecular dynamics simulations.

### Accelerated molecular dynamics simulations

Accelerated molecular dynamics (aMD) simulations utilize an enhanced sampling method that applies a bias or boost potential to the true potential that effectively raises the minima in the potential energy surface, leading to an enhanced escape rate and sampling of longer timescale events with shorter MD simulations. Using the Amber v14 software package (*Case et al., 2017*), conventional MD (cMD) simulations of PLC-γ1 and PLC-γ1(D1165H) in explicit solvent were used for equilibration, followed by a five nsec cMD simulation for calculating the boost potential, followed by completion of 145 nsec of boosted aMD simulations. The explicit solvent systems for PLC-γ1 and PLC-γ1(D1165H) were generated using LEaP and contained 1173 amino acid residues (PLC-γ1: 18,877 atoms, PLC-γ1(D1165H): 18,882), Na$^+$ ions for charge neutralization (PLC-γ1: 25 ions, PLC-γ1 (D1165H): 24 ions), and TIP3P water molecules in an octahedral box (PLC-γ1: 39,688 TIP3P, PLC-γ1 (D1165H): 39,700), for total system sizes of ~138,000 atoms. The ff14SB force field (*Maier et al., 2015*) was used for parameterization and simulations run using pmemd.cuda. First, the systems underwent minimization for 10,000 steps with a convergence criterion of 0.05 kcal/mol-Å. This was followed by 200 psec dynamics (NVT ensemble) for heating, with the thermostat target temperature increasing linearly from 0K to 300K over the course of the 200 psec using a Berendsen thermostat with a relaxation time of 0.5 psec. Protein atoms were restrained with a harmonic potential of weight 1.0 kcal/mol-Å$^2$. For relaxation and density equilibration, 300 psec dynamics (NPT ensemble) were completed using the Langevin thermostat with a collision frequency of 2.0 psec$^{-1}$ and isotropic pressure scaling with a relaxation time of 1.0 psec. At this stage, protein atoms were restrained with weight 0.1 kcal/mol-Å$^2$. During the next 500 psec of dynamics (NPT ensemble), there were no restraints placed on protein atoms. The last step of cMD was a 5 nsec simulation with snapshots

saved every 5 psec. All simulations were under periodic boundary conditions, with a 1 fsec time-step with hydrogen atoms constrained by SHAKE.

Boost potential parameters were determined from the average dihedral energy (Ed) and total potential energy (Ep) over the last 5 nsec of cMD according to the protocol described by *Pierce et al. (2012)*:

EthreshD = Ed + (four kcal mol$^{-1}$ residue$^{-1}$ * # solute residues)
alphaD = (0.2)*(four kcal mol$^{-1}$ residue$^{-1}$ * # solute residues)
EthreshP = Ep + (0.16 kcal mol$^{-1}$ atom$^{-1}$ * # atoms)
alphaP = (0.16 kcal mol$^{-1}$ atom$^{-1}$ * # atoms)

Using this formulation, the boost potential parameters were EthreshD = 19821, alphaD = 938.4, EthreshP=-395856, and alphaP = 22075. for PLC-γ1 aMD, and EthreshD = 19842, alphaD = 938.4, EthreshP=-395818, and alphaP = 22081 for PLC-γ1(D1165H) aMD.

Two independent simulations with the same starting conformation but different velocity distributions were completed for both PLC-γ1 and PLC-γ1(D1165H). Results from the analysis were consistent between each pair of replicate trajectories, and results from one representative trajectory for each protein are presented in *Figure 4*. Coordinate trajectories were processed using cpptraj for determining the average structure over the 75–150 nsec period of the simulation, where the conformation appeared to settle into a meta-stable state, as well as the correlated motions for Cα atoms during the first 75 nsec leading to the meta-stable state.

## Acknowledgements

We gratefully acknowledge Drs. M Lemmon and J Schlessinger for a critical reading of the manuscript. We thank Dr. M Miley for assistance with in-house collection of X-ray diffraction data and Dr. S Endo-Streeter for assistance with validation of the PLC-γ1 model. We thank the staff of SER-CAT beamlines 22-ID and 22-BM for assistance with collection of X-ray diffraction data; use of the Advanced Photon Source was supported by the U.S. Department of Energy, Office of Science, Office of Basic Energy Sciences, under Contract W-31–109-Eng-38. This work was supported by The National Institutes of Health Grants R01-GM057391 (JS) and R01-GM098894 (QZ and JS). ES-P was supported by a National Science Foundation Graduate Research Fellowship under Grant No. DGE-1650116. The UNC Macromolecular Crystallization Core Facility and R L Juliano Structural Bioinformatics Core Facility are supported by P30-CA016086.

## Additional information

### Competing interests

John Sondek: partial ownership of KXTbio, Inc which licenses the production of WH-15. The other authors declare that no competing interests exist.

### Funding

| Funder | Grant reference number | Author |
| --- | --- | --- |
| National Institutes of Health | R01-GM057391 | John Sondek |
| National Institutes of Health | R01-GM098894 | Qisheng Zhang John Sondek |
| National Science Foundation | DGE-1650116 | Edhriz Siraliev-Perez |

The funders had no role in study design, data collection and interpretation, or the decision to submit the work for publication.

### Author contributions

Nicole Hajicek, Formal analysis, Supervision, Investigation, Methodology, Writing - original draft, Writing - review and editing; Nicholas C Keith, Edhriz Siraliev-Perez, Formal analysis, Investigation, Writing - review and editing; Brenda RS Temple, Software, Formal analysis, Investigation,

Methodology, Writing - review and editing; Weigang Huang, Resources, Writing - review and editing, Synthesized and characterized WH-15; Qisheng Zhang, Resources, Supervision, Funding acquisition, Writing - review and editing; T Kendall Harden, Conceptualization, Methodology, Writing - original draft, Writing - review and editing; John Sondek, Conceptualization, Formal analysis, Supervision, Funding acquisition, Methodology, Writing - original draft, Writing - review and editing

## Author ORCIDs
Nicole Hajicek (iD) https://orcid.org/0000-0001-7457-4830
Nicholas C Keith (iD) https://orcid.org/0000-0002-4050-9261
Edhriz Siraliev-Perez (iD) https://orcid.org/0000-0003-1824-863X
Brenda RS Temple (iD) https://orcid.org/0000-0002-9233-0191
John Sondek (iD) https://orcid.org/0000-0002-1127-8310

## Decision letter and Author response
Decision letter https://doi.org/10.7554/eLife.51700.sa1
Author response https://doi.org/10.7554/eLife.51700.sa2

## Additional files

### Supplementary files
• Supplementary file 1. Crystallization statistics for PLC-γ1.

• Transparent reporting form

### Data availability
Coordinates and structure factors for PLC-gamma1 are deposited in the Protein Data Bank under accession number 6PBC.

The following dataset was generated:

| Author(s) | Year | Dataset title | Dataset URL | Database and Identifier |
|---|---|---|---|---|
| Hajicek N, Sondek J | 2019 | Structural basis for the activation of PLC-gamma isozymes by phosphorylation and cancer-associated mutations | https://www.rcsb.org/structure/6PBC | RCSB Protein Data Bank, 6PBC |

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
