## [Decision Letter]

**Acceptance summary:**

This article represents an important advance in our understanding of the normal regulation and the dysregulation of a key family of signaling enzymes, the phospholipase C-γ proteins. The structure of near-full-length auto-inhibited PLC-γ1, with concomitant biochemical analysis, reveals a clear picture of how PLC-γ isozymes are activated in response to various input signals. The structure and analysis also suggest how disease-associated mutations in PLC-γ proteins can enhance PLC-γ enzymatic activity and signaling. The results presented in this report will be of interest both to structural biologists and cancer biologists. Notably, as the authors point out in the article, the structural insights presented here should also lay the foundation for the development of PLC-γ targeted therapies.

**Decision letter after peer review:**

Thank you for submitting your article "Structural basis for the activation of PLC-γ isozymes by phosphorylation and cancer-associated mutations" for consideration by *eLife*. Your article has been reviewed favorably by three peer reviewers, including Neel Shah as the Reviewing Editor and Reviewer #1, and the evaluation has been overseen by John Kuriyan as the Senior Editor. The following individual involved in review of your submission has agreed to reveal their identity: Peter Gierschik (Reviewer #2).

The reviewers have discussed the reviews with one another and the Reviewing Editor has drafted this decision to help you prepare a revised submission.

Summary:

Hajicek et al. present the structure of full-length PLC-γ 1 in its auto-inhibited state. The structure shows how the multiple regulatory domains of the enzyme act in concert to prevent membrane localization and access to phospholipid substrates. Furthermore, the structure suggests how recruitment of an activating kinase, such as *FGFR1*, primes the molecule for the large conformational rearrangement that must occur to fully activate the enzyme. Their results also nicely explain how many disease associate mutations found in PLC-γ dysregulate the enzyme. The structure of auto-inhibited PLC-γ1 reveals that many disease-associated mutations lie at interdomain interfaces and likely disrupt auto-inhibitory interactions.

Overall this is a clear and well written manuscript that provides new mechanistic understanding of an important, and tightly regulated, signaling molecule. The work will be of interest to a wide audience.

Essential revisions:

The reviewers have agreed that no essential experimental revisions are required.

1) The presentation of the results aims at understanding a dynamic process, involving subdomain movements and reorientation of the enzyme vis-à-vis the plasma membrane containing its substrate. Since the structure presented here represents a single snapshot of this dynamic process, many aspects of this process are left open for future discoveries. Our recommendation, therefore, is to more clearly separate, at many points throughout the manuscript, experimental facts from still hypothetical processes (e.g. "wholesale rearrangement") that evolve from the snapshot scenario upon enzyme activation.

2) PLC-γ isozymes are phosphorylated at several tyrosine residues, e.g. PLC-γ2 at Y753, Y759, Y1197, and Y1217. The phosphorylation of these residues has been reported to be functionally important (JBC 276:38595-601 and 276:47982-92). Three of these residues are lacking in the structure analyzed here, but the authors should mention the importance of phosphorylation sites other than Y783 in PLC-γ1. Can anything about these other sites be inferred from this new structure?

3) The authors suggest that activation of PLC-γ isozymes requires wholesale rearrangement of the regulatory domains relative to the catalytic core, and indicate that the details of this change await further studies. In their 2012 Structure article, Bunney and co-workers examine structural rearrangements in the regulatory domains by SAXS and NMR (PMID: 23063561). The authors should address how those low-resolution models relate to the high-resolution structure and mechanistic models described in this manuscript.

4) In the cellular studies, highlighted in Figures 5 and 6, were phosphorylation states/levels of PLC-γ1 mutants measured (with a pTyr783-specific antibody)? If so, this data could be included to help interpret the mechanisms by which the different mutants respond to EGF stimulation.

5) For the data in Figure 6B, it would be interesting to know what the fold-change in activity is for each mutant in the +EGF condition relative to the -EGF condition. Does a comparison of fold-change upon EGFR stimulation across the mutants say anything about the mechanism of activation?

6) Related to the previous point, it is interesting that many of the mutants that destabilize the auto-inhibited state can achieve higher than wild-type levels of activity upon EGFR stimulation. This suggests that activation of PLC-γ1 by EGFR (presumably via phosphorylation of Tyr783) may not alter PLC-γ1 in the same way as these mutations. How do these mutants achieve higher activity in their fully-activated state? Can the authors comment on this?

7) The manuscript very nicely describes that there are differences in enzyme activity, depending on the presentation of the substrate in a water-soluble form, in micelles, in vesicles, and in intact cells. These differences are mostly explained with the intrinsic structural features of the enzyme described in this work. However, in intact cells, extrinsic factor could come into play, such as upstream regulators of the enzyme, which might be present and active, at least to some degree, in transfected cultured model cells.

8) Sequence and structural comparison of the C2 domains of PLC-δ1 and PLC-γ2 suggest that the D1165MFS sequence (PLC-γ1 numbering) may come close to the Ca^2+^ binding site at the tip of the domain. This, together with the fact that this region of PLC-γ2 has previously been linked to the Ca^2+^ "off" response in B cells (Nishida et al., 2003) should be considered and dealt with in the text. Also, PLC-δ4 is divergent from PLC-δ1 in that region, with functional consequences (JBC 277:3568-75).

9) Many of the point mutations in PLC-γ2 cause BTK inhibitor resistance in CLL, but have not been described to be (causally) linked to cancer. Hence, the statement, that "… most of the substitutions and small deletions in these isozymes" (i.e. in both isozymes) that are linked to cancer" has to be restricted to PLC-γ1.

10) Regarding the accelerated MD simulations: It is not clear how many independent trajectories were run and analyzed to assess flexibility and correlated motions. Could the authors indicate in the Materials and methods section and/or figure legend the number and duration of independent trajectories?

11) The "Δ25" mutant is defined somewhat cryptically in the Figure 1 legend but nowhere else. It would be helpful if this nomenclature was defined in the main text, where the crystallographic construct is first described.

12) Some of the PLC-γ1 point mutants analyzed are direct counterparts of PLC-γ2 mutants found in tumor cells from ibrutinib-resistant patients. This should be referred to in the text wherever appropriate.

---

## [Author Response]

Essential revisions:The reviewers have agreed that no essential experimental revisions are required.1) The presentation of the results aims at understanding a dynamic process, involving subdomain movements and reorientation of the enzyme vis-à-vis the plasma membrane containing its substrate. Since the structure presented here represents a single snapshot of this dynamic process, many aspects of this process are left open for future discoveries. Our recommendation, therefore, is to more clearly separate, at many points throughout the manuscript, experimental facts from still hypothetical processes (e.g. "wholesale rearrangement") that evolve from the snapshot scenario upon enzyme activation.

We have updated the main text in several places, as well as the legend to Figure 1, to better emphasize which aspects of the activation process are still hypothetical.

2) PLC-γ isozymes are phosphorylated at several tyrosine residues, e.g. PLC-γ2 at Y753, Y759, Y1197, and Y1217. The phosphorylation of these residues has been reported to be functionally important (JBC 276:38595-601 and 276:47982-92). Three of these residues are lacking in the structure analyzed here, but the authors should mention the importance of phosphorylation sites other than Y783 in PLC-γ1. Can anything about these other sites be inferred from this new structure?

We have added an additional paragraph to the text highlighting the existence and functional implications of phosphorylation sites besides Tyr783 in PLC-γ1 and Tyr759 in PLC-γ2. To the best of our knowledge, only Tyr775 in PLC-γ1 has also been demonstrated to regulate enzyme activity; this regulation seems to be most relevant in the context of soluble tyrosine kinases coupled to immune receptors. The equivalent site in PLC-γ2, Tyr753, has a similar regulatory role. We note that phosphorylation of PLC-γ2 on Tyr1197 and Tyr 1217 has been reported; these sites are not conserved in PLC-γ1. Unfortunately, the Δ25 deletion encompasses Tyr775, limiting the conclusions that can be drawn about how this site regulates activity.

3) The authors suggest that activation of PLC-γ isozymes requires wholesale rearrangement of the regulatory domains relative to the catalytic core, and indicate that the details of this change await further studies. In their 2012 Structure article, Bunney and co-workers examine structural rearrangements in the regulatory domains by SAXS and NMR (PMID: 23063561). The authors should address how those low-resolution models relate to the high-resolution structure and mechanistic models described in this manuscript.

With respect to the structural models and activation mechanism proposed by Bunney et al. (PMID: 23063561), a paragraph has been added to the text discussing similarities and differences with the crystal structure.

4) In the cellular studies, highlighted in Figures 5 and 6, were phosphorylation states/levels of PLC-γ1 mutants measured (with a pTyr783-specific antibody)? If so, this data could be included to help interpret the mechanisms by which the different mutants respond to EGF stimulation.

Excellent question, however, the phosphorylation state of PLC-γ1 mutants was not assessed in this study.

5) For the data in Figure 6B, it would be interesting to know what the fold-change in activity is for each mutant in the +EGF condition relative to the -EGF condition. Does a comparison of fold-change upon EGFR stimulation across the mutants say anything about the mechanism of activation?

Figure 6 has been modified to indicate the fold-change in activity between the -EGF and +EGF conditions. It is unclear if there is any relationship between fold-changes in EGFR stimulation and mechanisms of activation.

6) Related to the previous point, it is interesting that many of the mutants that destabilize the auto-inhibited state can achieve higher than wild-type levels of activity upon EGFR stimulation. This suggests that activation of PLC-γ1 by EGFR (presumably via phosphorylation of Tyr783) may not alter PLC-γ1 in the same way as these mutations. How do these mutants achieve higher activity in their fully-activated state? Can the authors comment on this?

We agree that the higher levels of receptor-dependent lipase activity displayed by mutant forms of PLC-γ1 is intriguing; furthermore, we note in the Discussion that mutant versions of PLC-γ2 are reported to behave similarly. That is, in the context of receptor stimulation, cells expressing mutated versions of PLC-γ2 achieve higher levels of calcium mobilization (and those calcium levels are sustained for a longer period of time) than cells expressing wild-type PLC-γ2 (PMID: 24869598). How mutations might drive supra-activation of the PLC-γ isozymes is not immediately clear, although it may reflect a combination of hypersensitivity to upstream activators and a slower rate of return to the autoinhibited state. We suggest that cancer-associated substitutions drive the equilibrium away from the autoinhibited state towards more active conformations, effectively increasing the amount of time the lipase spends in active state. These ideas are further elaborated in the Discussion.

7) The manuscript very nicely describes that there are differences in enzyme activity, depending on the presentation of the substrate in a water-soluble form, in micelles, in vesicles, and in intact cells. These differences are mostly explained with the intrinsic structural features of the enzyme described in this work. However, in intact cells, extrinsic factor could come into play, such as upstream regulators of the enzyme, which might be present and active, at least to some degree, in transfected cultured model cells.

We agree that extrinsic cellular factors may impact regulation of PLC-γ1. We believe this point is implicitly addressed in the Discussion and has not been developed further.

8) Sequence and structural comparison of the C2 domains of PLC-δ1 and PLC-γ2 suggest that the D1165MFS sequence (PLC-γ1 numbering) may come close to the Ca^2+^ binding site at the tip of the domain. This, together with the fact that this region of PLC-γ2 has previously been linked to the Ca^2+^ "off" response in B cells (Nishida et al., 2003) should be considered and dealt with in the text. Also, PLC-δ4 is divergent from PLC-δ1 in that region, with functional consequences (JBC 277:3568-75).

We have expanded the Discussion on the potential of the PLC-γ1 C2 domain to ligate calcium to include the fact that the corresponding region in PLC-γ2 has been implicated in Ca^2+^-dependent translocation with functional implications for Ca^2+^ signaling in B lymphocytes.

9) Many of the point mutations in PLC-γ2 cause BTK inhibitor resistance in CLL, but have not been described to be (causally) linked to cancer. Hence, the statement, that "… most of the substitutions and small deletions in these isozymes" (i.e. in both isozymes) that are linked to cancer" has to be restricted to PLC-γ1.

Additional references that link PLC-γ2 to cancer were added.

10) Regarding the accelerated MD simulations: It is not clear how many independent trajectories were run and analyzed to assess flexibility and correlated motions. Could the authors indicate in the Materials and methods section and/or figure legend the number and duration of independent trajectories?

The Materials and methods section has been updated with the requested information.

11) The "Δ25" mutant is defined somewhat cryptically in the Figure 1 legend but nowhere else. It would be helpful if this nomenclature was defined in the main text, where the crystallographic construct is first described.

The °25 mutation is now defined in the main text and readers are directed to the Materials and methods section for a detailed description of the deletion. Figure 1—figure supplement 1 was also modified to highlight explicitly the Δ25 deletion.

12) Some of the PLC-γ1 point mutants analyzed are direct counterparts of PLC-γ2 mutants found in tumor cells from ibrutinib-resistant patients. This should be referred to in the text wherever appropriate.

The direct correspondence between the PLC-γ1(D1165H)/PLC-γ2(D1140G) substitutions and the PLC-γ1(R687W)/PLC-γ2(R665W) substitutions is noted at the first mention of PLC-γ1(D1165H) and PLC-γ1(R687W) in the text.